# ONLINE REUSABLE RESOURCE ALLOCATION WITH ADVERSARIAL REQUESTS

## ABSTRACT

We study an online reusable resource allocation problem with adversarial inputs, where a platform must decide, over a horizon $T$, whether to accept incoming job requests with adversarial resource demands and durations. The goal is to maximize cumulative revenue subject to resource budget constraints. We propose a class of *online dual dynamic learning algorithms* and *learning from pricing experts algorithms* that achieve asymptotically optimal competitive ratios, remain computationally efficient, and further improve performances across different regimes of maximum job duration and resource demand. We further extend the model to allow flexible resource allocations, where neither the demand nor the duration is fixed; instead, the amount of allocated resource influences job duration. To address this more general setting, we reduce the problem to the binary allocation case via resource discretization. We prove that the resulting loss is bounded by a constant depending only on the total budget, and this bound is nearly optimal.

## 1 INTRODUCTION

Resource allocation is a fundamental problem in operations research, with effective solutions being critical to a wide range of applications, including revenue management and supply chain management (Karp et al., 1990; Mehta et al., 2013; Golrezaei et al., 2014; Aouad & Saban, 2023; Owen & Simchi-Levi, 2018). In particular, allocation involving reusable resources presents additional challenges, as the constraints are more complex and generalize the non-reusable case. Examples of reusable resources include hotel rooms, shared laboratory equipment, and, with the rapid development of modern Machine Learning and Artificial Intelligence, cloud computing resources and GPUs. The growing demand for GPU rental and inference tasks further underscores the importance of developing efficient allocation methods for reusable resources.

Over the years, extensive research has studied resource allocation under various settings, including stochastic, adversarial, and non-stationary environments (Balseiro et al., 2023; Devanur et al., 2019), as well as scenarios with bounded resource requests and large capacity/inventory regimes (Goyal et al., 2025; Gong et al., 2022; Lei & Jasin, 2020). However, the study of *adversarial* resource allocation for reusable resources remains largely unexplored. In this work, we investigate the problem of online reusable resource allocation in adversarial environments, where requests arrive sequentially with associated revenues, and no statistical assumptions are made on the request sequence. For each incoming job, the platform must determine an action: either reject the job, or accept it with a fixed resource and duration requirement. The objective is to design an effective scheduling and allocation strategy that maximizes the cumulative revenue over time.

**Our results and techniques.** In the online reusable resource allocation problem with adversarial requests, at time $t$ the platform observes the requested resource $x_t$, the associated revenue $f_t(x_t)$, and the usage duration $\lambda_t(x_t)$. It must then decide whether to accept or reject the request while ensuring that the total active allocation at any time does not exceed the budget $B$. The adversarial input setting significantly increases the difficulty of the problem. If the platform accepts requests too aggressively, subsequent requests may offer substantially larger revenues, which cannot be realized due to insufficient remaining resources. Conversely, if the platform adopts an overly conservative strategy and rejects early requests, the following requests may provide only small revenues. More-

over, long-duration requests are particularly challenging, as they constrain the available resources for many future rounds.

To address these challenges, we design two classes of algorithms. The first class is based on a *primal–dual framework*, in which updates are performed in the dual space. Since the platform's allocation decisions are temporally coupled by the reusable resource constraints, we employ a Lagrangian formulation to decouple the decisions across time steps. Over a horizon of length $T$, the reusable resource constraint induces $T$ dual variables, of which only a subset is active and effective at time $t$. We aggregate all dual variables effective at time $t$ to form a new dual variable $\nu_t$. This aggregated variable $\nu_t$ naturally serves as a *resource price* at time $t$, providing a threshold for determining whether a job request should be accepted. However, because job durations vary, the number of active dual variables changes over time, making the optimal $\nu_t^\star$ inherently dynamic. In contrast to the non-reusable case—where a single static dual variable $\nu^\star$ suffices—we employ a dynamic online learning approach to adaptively learn and update the evolving sequence $\{\nu_t\}_{t=1}^T$ in Theorem 2 and Theorem 3.

Using our dual online allocation algorithm, our Theorem 1 recovers the optimal asymptotic competitive ratio $Tx_{\max}/B$ (Balseiro & Gur, 2019) in the degenerate non-reusable case ($\lambda_t = T$ or $\lambda_t \to \infty$), where $x_{\max} = \max_t x_t$. But this algorithm turns out to be too conservative when the resource durations are much less than $T$. So we design a new algorithm parametrized by the *manual maximum duration* $\lambda_{\max}$, where $\lambda_{\max} \geq \overline{\lambda} = \max_t \lambda_t$. The exact value of $\overline{\lambda}$ is not required—any upper bound suffices—and any reasonable estimate yields a proportionally good performance guarantee. We applies an alternating block technique, which partition the whole $T$ horizon into $\lceil \frac{T}{\lambda_{\max}} \rceil$ blocks, each block containing a time period of length $\lambda_{\max}$. And we work only on even or odd intervals uniformly at random. This technical trick ensures that the resource is fully replenished at the beginning of each interval that we choose to work on. By this means, we obtain a much tighter competitive ratio $\lambda_{\max}x_{\max}/B$. And there is a trade-off in selecting the parameter $\lambda_{\max}$. Conceptually, a bigger (smaller) $\lambda_{\max}$ leads to a more aggressive (conservative) allocation strategy. Technically, a smaller $\lambda_{\max}$ leads to a better competitive ratio, but a worse regret bound.

In the dual dynamic online allocation framework, the dual variable serves as a resource price based only on the resource amount consumed, without incorporating duration. This limitation motivates the alternating block technique, which is particularly effective when $\overline{\lambda}$ is super-constant. For the constant duration case, we develop a *duration-adapted online dynamic allocation algorithm* that incorporates the current job duration $\lambda_t$ directly into the gradient of the dual function. This eliminates the need for alternating blocks. In this algorithm, the dual variable $\nu_t$ serves as a price per usage volume, and a request is accepted whenever $\frac{f_t}{\lambda_t x_t} \geq \nu_t$. The algorithm achieves an asymptotic competitive ratio of $3\lambda_{\max}x_{\max}/B$, nearly as strong as that of the alternating block algorithm but without the reliance on an initial random choice.

Finally, we propose a complementary algorithm based on *pricing experts*. Here, we model every exper as a two-sided pricing threshold: each expert $k$ corresponds to an interval $[\phi_k, \phi_{k+1})$. At time $t$, a request is accepted if its revenue per usage unit, $\frac{f_t}{\lambda_t x_t}$, lies in the selected expert's interval; otherwise, it is rejected. We use alternating blocks and run *learning from expert advice* algorithm, such as *Hedge* (Littlestone & Warmuth, 1994; Cesa-Bianchi & Lugosi, 2006) to select an expert at each active block. This algorithm achieves an asymptotic competitive ratio $\frac{2H(B+(2\lambda_{\max}-1)x_{\max})}{B-x_{\max}}$, which can outperform the primal–dual guarantee when $x_{\max}$ is large and $\overline{\lambda}$ is constant. The key advantage is that, unlike primal–dual algorithms, which require $\lambda_{\max}$ to be sublinear in $T$ to ensure sublinear regret, the pricing-expert approach allows setting $\lambda_{\max} = \overline{\lambda}$ exactly.

We next consider an intricate and challenging flexible resource allocation setting, in which the platform can determine not only whether to accept a job, but also how much resource to allocate at each round. In this model, each job request has a fixed revenue, but its duration depends on the amount of resources allocated, introducing additional complexity. As a motivating example, consider LLM inference tasks arriving in a stream. The platform needs to allocate different amounts of GPUs to each task. We will *eventually* finish a task even if we allocate only a few GPUs. However, the more GPUs we allocate, the faster we can finish a task, which means all these GPUs can be freed up earlier for future tasks. To tackle this problem, we develop a binary reduction that transforms this more general allocation problem into the binary allocation problem discussed above. The reduction proceeds by

discretizing the interval $[0, B]$ into powers of two, and the platform chooses one of them uniformly at random as the resource allocation budget per round. We further prove that the reduction loss in a ratio form is bounded by $\frac{1}{2(1+\lceil \log x_{\max} \rceil)}$, which only depends on the maximum required resource amount $x_{\max}$. And we prove that our reduction is nearly optimal by giving a matching lower bound on any reduction.

## 2 RELATED WORK

The online allocation problem has been extensively studied in operations research and computer science, with applications in revenue management, cloud resource provisioning, and online task scheduling. A large body of work focuses on the non-reusable resource setting (Feldman et al., 2009; Jasin, 2014; Devanur et al., 2019; Balseiro et al., 2023; Agrawal et al., 2014). In the classical model of online bipartite matching with adversarial arrival, Karp et al. (1990) showed that a simple greedy algorithm achieves a $1/2$-competitive ratio, and that this can be improved to $1 - 1/e$ in the random-order model using the Ranking algorithm. Building on this, seminal works such as Mehta et al. (2007) and Buchbinder et al. (2007) study the AdWords problem, a generalized online matching problem that incorporates heterogeneous initial resource inventories and requests with multi-unit demands. A related stream in revenue management and online knapsack studies LP-based "re-solving" heuristics against a deterministic LP benchmark: Jasin & Kumar (2012) show that frequently re-solving the LP enables constant regret under a non-degeneracy condition, while Bumpensanti & Wang (2020) design an infrequent re-solving policy with uniformly bounded loss without requiring non-degeneracy. More recent work develops general constant-regret frameworks for stochastic online allocation and packing (Vera & Banerjee, 2019; Arlotto & Gurvich, 2019). Vera et al. (2021) and Banerjee & Freund (2020) use Bellman-inequality or uniform loss to design policies that repeatedly solve LP relaxations and achieve regret that does not grow with the horizon or capacities.

The primal–dual method has been particularly effective for revenue optimization under budget constraints. For example, Agrawal et al. (2014); Kesselheim et al. (2014); Devanur et al. (2019) design primal–dual algorithms that dynamically update threshold prices or allocation decisions, often requiring the solution of a linear program at each round or period. Gupta & Molinaro (2016) extend the results to the random permutation model. Balseiro et al. (2023) propose a dual mirror descent algorithm that adapts across different environments and achieves a best-of-many-worlds guarantee. We extend their adversarial input model to the reusable resource setting, noting that the non-reusable case is a special instance of the reusable case. We also show that our results match the known lower bounds when the reusable setting reduces to the non-reusable one (Balseiro & Gur, 2019). Yang et al. (2024) study online allocation with replenishable budgets using similar primal–dual methods to Balseiro et al. (2023). In their framework, replenishment is oblivious: the decision-maker's actions do not affect the future replenishment. By contrast, in the reusable resource setting, the decision maker's actions directly influence future resource availability, making replenishment adaptive. Overall, these approaches rely critically on the non-reusable resource assumption, and their analyses depend on the associated non-adaptive budget constraints; therefore, these techniques can not be applied directly to the reusable resource case.

In contrast, online reusable resource allocation remains less explored. Levi & Radovanović (2010) consider an admission control problem and a pricing problem with reusable resource and propose a class selection policy that is asymptotically optimal as the capacity of the system grows to infinity. Chen et al. (2017) study a similar problem with advanced reservation. There is a line of work on reusable resource allocation under stochastic assumptions about arrivals (Dickerson et al., 2021; Feng et al., 2022; Zhang & Cheung, 2022; Rusmevichientong et al., 2020; Feldman et al., 2009; Kanoria & Qian, 2024). For example, Feng et al. (2022) assume that request types are drawn independently from known heterogeneous distributions over a finite horizon, while Zhang & Cheung (2022) assume i.i.d. requests from a common finite-support distribution. Goyal et al. (2025) and Gong et al. (2022) investigate online bipartite matching with reusable resource and online assortment optimization, respectively, under the assumptions that service durations depend on the resource but not on the job, and that each match consumes exactly one unit of resource. In addition, Goyal et al. (2025) focus on the large-capacity regime. Similar assumptions appear in several other works (Rusmevichientong et al., 2020; Lei & Jasin, 2020; Besbes et al., 2019). Their analyses rely heavily on these assumptions, leaving the general adversarial input setting largely unexplored. Our work

addresses this gap by studying reusable resource allocation under general adversarial inputs, where request sizes, usage durations, and total budgets are arbitrary or subject only to mild assumptions.

# 3 PROBLEM STATEMENT AND PRELIMINARY

**Problem setup.** We study a reusable resource allocation problem with a single resource type over a time horizon of $T$ rounds. At each round $t$, a job request $(f_t, x_t, \lambda_t)$ will arrive, and this request information is revealed to the platform before the platform makes the decision. $x_t$ is the amount of resource demand of job request $t$, and $x_t \in [x_{\min}, x_{\max}]$. The platform must decide immediately whether to accept the request or not upon receiving the request. Let $a_t \in \{0, 1\}$ denote this binary decision: $a_t = 1$ if the job is accepted, and $a_t = 0$ otherwise. If accepted, the job generates reward $f_t(x_t) \in [f_{\min}, f_{\max}]$ and occupies resources for $\lambda_t(x_t) \geq 1$ rounds. We use the shorthand notation $(f_t, x_t, \lambda_t) \equiv (f_t(x_t), x_t, \lambda_t(x_t))$ when there is no ambiguity. We assume the duration $\lambda_t$ at time $t$ is a positive integer without loss of generality, since we can let the duration be $\lceil \lambda_t \rceil$ when $\lambda_t$ is not an integer. The platform has $B$ reusable resource in total. In this paper, we consider the adversarial input model, which means we do not make any statistical assumption on the request.

Since jobs may last for multiple rounds, we must ensure that the total active resource at any time does not exceed $B$. Specifically, job $\tau$ is active at time $t$ if $\lambda_\tau(x_\tau) \geq t - \tau + 1$, i.e., its duration covers round $t$. We define the feasible allocation set $\mathcal{A} := \{a_t\}_{t=1}^T$ as those satisfying $\sum_{\tau=1}^t a_\tau \cdot \mathbb{I}\{\lambda_\tau(x_\tau) \geq t - \tau + 1\} \cdot x_\tau \leq B$ for every $t \in [T]$. The goal is to select, in the online fashion described above, $\mathcal{A} := \{a_t\}_{t=1}^T$ that maximizes:

$$\text{OPT}_b(\mathcal{A}^\star) := \max_{\mathcal{A}} \sum_{t=1}^T a_t f_t, \quad \text{s.t.} \sum_{\tau=1}^t a_\tau \cdot \mathbb{I}\{\lambda_\tau(x_\tau) \geq t - \tau + 1\} \cdot x_\tau \leq B, \ \forall t. \tag{1}$$

Let $\mathcal{A}^\star = \{a_t^\star\}_{t=1}^T$ be the optimal binary schedule achieving $\text{OPT}_b$.

**The primal-dual framework.** The structure of the constraints reflects a temporal coupling of jobs due to their durations, which poses unique algorithmic challenges compared to classical knapsack or scheduling problems. An approach to deal with this problem is to introduce Lagrange multipliers as dual variables. The use of Lagrange multipliers and dual prices to relax capacity constraints is folklore in online allocation. We use this template for reusable resources and instantiate three new formulations that map to our algorithms. For $\mu_t \geq 0, t \in [T]$, the Lagrangian of this optimization problem is:

$$\mathcal{L}(\mathcal{X}, \boldsymbol{\mu}) = \sum_{t=1}^T a_t f_t(x_t) - \mu_t \left( \sum_{\tau=1}^t a_\tau \cdot \mathbb{I}\{\lambda_\tau(x_\tau) \geq t - \tau + 1\} \cdot x_\tau - B \right) \tag{2}$$

$$= \sum_{t=1}^T a_t \left( f_t(x_t) - x_t \sum_{\tau=t}^{t+\lambda_t-1} \mu_\tau \right) + \frac{B}{T} \sum_{t=1}^T \mu_t \tag{3}$$

$$= \sum_{t=1}^T a_t \left( f_t(x_t) - x_t \nu_t \right) + \frac{B}{\lambda_{\max}} \left( \sum_{\tau=t}^{t+\lambda_{\max}-1} \mu_\tau \right) + U_{\text{remain}} \tag{4}$$

$$= \sum_{t=1}^T a_t \left( f_t(x_t) - x_t \lambda_t \overline{\nu}_t \right) + \lambda_t \frac{B}{\lambda_{\max}} \left( \frac{1}{\lambda_t} \sum_{\tau=t}^{t+\lambda_{\max}-1} \mu_\tau \right) + U_{\text{remain}}. \tag{5}$$

We aggregate all dual variables effective at time $t$ into a new dual variable $\nu_t = \sum_{\tau=t}^{t+\lambda_t-1} \mu_\tau$ or $\overline{\nu}_t = \frac{1}{\lambda_t} \sum_{\tau=t}^{t+\lambda_t-1} \mu_\tau$, which we then learn and update over time. This aggregated variable $\nu_t$ provides a threshold for deciding whether a job request should be accepted, and we can interpret it as the *resource price* at time $t$. Because request durations vary, the number of active dual variables fluctuates across time, making the sequence of dual variable $\{\nu_t\}_{t=1}^T$ inherently dynamic. In contrast to the non-reusable case—where a single static dual variable $\nu^\star$ suffices—we employ a dynamic online learning approach to adaptively learn and update the dynamic sequence $\{\nu_t\}_{t=1}^T$.

By weak duality, the dual formulation provides an upper bound on the primal problem and offers a natural way to decompose the allocation problem across time, enabling time-dependent decision-

making. Based on this intuition, we solve the dual problem $D(\boldsymbol{\nu}) = \max_{\mathcal{A}} \mathcal{L}(\mathcal{X}, \boldsymbol{\nu})$ by iteratively updating the dual variables $\nu_t$ to minimize $D(\boldsymbol{\nu})$ and using them to guide allocation. Concretely, at each round $t$, we accept a request if its revenue exceeds the threshold, $f_t(x_t) - x_t \nu_t \geq 0$, and reject it otherwise.

Note that the Lagrangian in Eq. (2) can be expressed in different formulations depending on the information available about job durations.

- **Unknown durations.** When no information is available and the maximum duration can be as large as $T$, we adopt the conservative formulation in Eq. (3) to design our *dual online allocation algorithm*. In this case, we set the average budget across time as $\rho = \frac{B}{T}$.
- **Bounded durations.** When the maximum duration is known to satisfy $\max_t \lambda_t = \overline{\lambda} \leq \lambda_{\max}$, the Lagrangian can be reformulated in alternative ways. Here, $\lambda_{\max}$ is treated as a manual set parameter of the algorithm and need not equal the exact maximum duration. $\lambda_{\max}$ only needs to be a rough upper bound on the true $\overline{\lambda} = \max_{t \in [T]} \lambda_t$. And in practice, batch and cloud schedulers routinely require each job to declare a maximum runtime at submission, and jobs that exceed this limit are rejected or terminated. This allows us to set a more aggressive average budget, $\rho = \frac{B}{\lambda_{\max}}$. Based on this formulation, we design two algorithms:
    1. the *alternating-block dual dynamic online allocation algorithm*, derived from Eq. (4);
    2. the *duration-adapted dual dynamic online allocation algorithm*, derived from Eq. (5), which uses the duration-average dual variable $\overline{\nu}_t = \frac{1}{\lambda_t} \sum_{\tau=t}^{t+\lambda_t-1} \mu_\tau$.

We exclude trivial cases from our algorithm: if $f_t = 0$, the request is always rejected, and if $\lambda_t = 1$, the request is always accepted. Several methods require divisibility conditions. In such cases, we omit boundary cases in the main paper for simplicity and clarity, but they are carefully handled in our analysis.

## 4 PRIMAL-DUAL BASED ALGORITHMS

We first present our primal-dual based algorithms, which generally perform better when $x_{\max}$ is moderate and only coarse information about durations is available, e.g., no bound or only an upper bound $\lambda_{\max}$ on $\overline{\lambda}$.

### 4.1 DUAL ONLINE ALLOCATION ALGORITHM

---

**Algorithm 1** Dual Online Allocation Algorithm

1: **Input:** Time horizon $T$, total resource $B$, initial dual variable $\nu_1 \geq 0$, and average resource budget $\rho$.
2: **for** $t = 1, \ldots, T$ **do**
3:     Receive request $(f_t, x_t, \lambda_t)$.
4:     Make the primal decision $a_t$:   $\widetilde{a}_t = \text{argmax}_{a \in \{0,1\}}\, a \cdot (f_t(x_t) - \nu_t x_t)$

$$\widetilde{B}_t = \sum_{\tau=1}^{t-1} a_\tau \cdot \mathbb{I}\{\lambda_\tau(x_\tau) \geq t - \tau + 1\} \cdot x_\tau + \widetilde{a}_t \cdot x_t, \quad a_t = \begin{cases} \widetilde{a}_t & \text{if } \widetilde{B}_t \leq B, \\ 0 & \text{otherwise.} \end{cases}$$

5:     Get the loss function and gradient of the dual variable

$$w_t(\nu_t) = \nu_t g_t = \begin{cases} \nu_t(\rho - a_t x_t), & \widetilde{B}_t \leq B, \\ 0, & \text{otherwise.} \end{cases}, \quad g_t = \begin{cases} \rho - a_t x_t, & \widetilde{B}_t \leq B, \\ 0, & \text{otherwise.} \end{cases}$$

6:     Update the dual variable by subroutine online learning algorithm:

$$\nu_{t+1} = \text{OL-ALG}(\nu_t, w_t(\nu_t))$$

7: **end for**

---

Suppose the platform does not know any information about the duration lengths. Motivated by the Lagrangian formulation Eq. (3), we first design a conservative algorithm for the online allocation

problem, setting the average resource budget as $\rho = \frac{B}{T}$. Our algorithm calls a subroutine, OL-ALG, in a blackbox way. We will discuss the choice of OL-ALG later.

**Theorem 1.** *For the online reusable resource allocation problem with adversarial requests* $\{f_t, x_t, \lambda_t\}_{t=1}^T$, *applying Algorithm 1 with parameters* $\nu_1 \geq 0$ *and* $\rho = \frac{B}{T}$, *we have for any* $T \geq 1$,

$$\frac{1}{\alpha}\text{OPT}_b\big(\{a_t^\star\}_{t=1}^T\big) - \sum_{t=1}^T a_t f_t(x_t) \leq f_{\max} + R(T, \nu),$$

*where* $\alpha = \max\{1, \frac{x_{\max}}{\rho}\}$, *and* $R(T, \nu)$ *is the regret of the subroutine* OL $-$ ALG, *satisfying* $\sum_{t=1}^T w_t(\nu_t) - \sum_{t=1}^T w_t(\nu) \leq R(T, \nu)$.

As a concrete example, if we choose the classical *online subgradient descent* (Zinkevich, 2003) or dynamic online learning algorithm *Ader* (Zhang et al., 2018a) as the subroutine OL-ALG, the regret satisfies $\max_\nu R(T, \nu) \leq O(GD\sqrt{T})$, where $\nu \in [0, f_{\max}/x_{\min}], G = \max\{\rho, x_{\max} - \rho\}, D = \frac{f_{\max}}{x_{\min}}$. Theorem 1 therefore shows that Algorithm 1 with $\rho = \frac{B}{T}$ is $\alpha$-asymptotic competitive; that is, $\frac{1}{T}\sum_{t=1}^T a_t f_t(x_t)$ is at least a $\frac{1}{\alpha}$-fraction of the average revenue of the offline optimum when $T \to \infty$. A smaller value of $\alpha$ indicates a stronger (better) competitive ratio. Since the duration function can be chosen arbitrarily, an adversary could set $\lambda_t = T$ for all $t \in [T]$, in which case the problem reduces to the online non-reusable resource allocation setting. As shown by Balseiro & Gur (2019), no algorithm can achieve a competitive ratio better than $\alpha = \frac{x_{\max}}{(B/T)}$ in this setting. Thus, our result is asymptotically optimal and matches the lower bound.

## 4.2 ALTERNATING-BLOCK DUAL DYNAMIC ONLINE ALLOCATION

When the maximum duration $\overline{\lambda} = \max_t \lambda_t$ is known, the algorithm can be refined to achieve improved competitive ratios. We partition the horizon $T$ into consecutive blocks, each of length $\lambda_{\max} \geq \lambda_t$, and represent each round as

$$t = (i-1) \cdot \lambda_{\max} + j, \quad j \in [\lambda_{\max}], \quad i \in \{1, 2, \ldots, \lceil T/\lambda_{\max} \rceil\}. \tag{6}$$

This block decomposition provides a structured scheduling framework that is well-suited to reusable resources under adversarial inputs. We only run our algorithm on odd or even blocks. Concretely, we adopt an alternating-block strategy: at the beginning of the allocation process, we uniformly sample $q \in \{0, 1\}$. The algorithm is then run only on the rounds belonging to *active blocks* of the form

$$t = (i-1) \cdot \lambda_{\max} + j, \quad j \in [\lambda_{\max}], \quad i \equiv q \pmod{2}, \tag{7}$$

with all tasks in the remaining blocks rejected to allow for full budget recovery. Consequently, the budget is restored to $B$ at the start of each active block.

Since the maximum duration is bounded by $\overline{\lambda}$, combined with the alternating block technique and inspired by the Lagrangian formulation Eq. (4), we can set the average budget more aggressively as $\rho = B/\lambda_{\max}$ instead of the conservative $\rho = B/T$. Recall that in Algorithm 1, the gradient term is $g_t = \rho - a_t x_t$. A larger value of $\rho$ increases the gradient and decreases the future price $\nu_{t+1}$, making the platform more inclined to accept incoming requests. Using $\rho = B/\lambda_{\max}$ avoids the overly pessimistic allocation induced by $\rho = B/T$ and yields a competitive-ratio bound of $2\lambda_{\max}x_{\max}/B$, which strictly improves upon the previous guarantee $Tx_{\max}/B$. For the purpose of dynamic regret analysis, we define the path length of the comparator sequence $\{\nu_t^\star\}_{t=1}^T$ as

$$P_T(\nu_1^\star, \ldots, \nu_T^\star) = \sum_{t=2}^T \big|\nu_t^\star - \nu_{t-1}^\star\big|.$$

This notion, introduced by Zinkevich (2003) and widely used in the online dynamic learning literature (Zhang et al., 2018b;a), quantifies the degree of fluctuation in the comparator sequence. A larger $P_T$ indicates greater variability, making the problem more challenging.

**Theorem 2.** *For the online reusable resource allocation problem with adversarial requests* $\{f_t, x_t, \lambda_t\}_{t=1}^T$, *if* $\overline{\lambda} = \max_t \lambda_t \leq \lambda_{\max}$, *then using the alternating block with Algorithm 1, parameters* $\nu_1 \geq 0$ *and* $\rho = B/\lambda_{\max}$, *we have for any* $T \geq 1$,

$$\frac{1}{2\alpha}\text{OPT}_b(\{a_t^\star\}_{t\in[T]}) - \mathbb{E}\left[\sum_{t=1}^T a_t f_t(x_t)\right] \leq \beta f_{\max} + R\big(T, P_T(\{\nu_t^\star\}_{t\in[T]})\big),$$

*with $\beta = \lceil T/\lambda_{\max} \rceil$, $\alpha = \max\{1, \frac{x_{\max}}{\rho}\}$, and $P_T(\{\nu_t^\star\}_{t\in[T]}) \leq \frac{\beta f_{\max}}{x_{\max}} \leq O\left(\frac{T}{\lambda_{\max}}\right)$. And $R\big(T, P_T(\{\nu_t^\star\}_{t\in[T]})\big)$ is the dynamic regret of the subroutine* $\mathrm{OL-ALG}$, *satisfying* $\sum_{t=1}^T w_t(\nu_t) - \sum_{t=1}^T w_t(\nu_t^\star) \leq R\big(T, P_T(\{\nu_t^\star\}_{t\in[T]})\big)$.

*Moreover, if we assume $x_{\max} \leq B/2$ and set $\rho = B/(2\lambda_{\max})$, then*

$$\frac{1}{2\alpha}\mathrm{OPT}_b(\{a_t^\star\}_{t\in[T]}) - \mathbb{E}\left[\sum_{t=1}^T a_t f_t(x_t)\right] \leq R\big(T, P_T(\{\nu_t^\star\}_{t\in[T]})\big).$$

For the subroutine dynamic learning algorithm OL-ALG, we can use the *Ader algorithm* (Zhang et al., 2018a), which satisfies $R(T, P_T) \leq O\big(GD\sqrt{T(1+P_T)}\big)$, where $G = \max\{\rho, x_{\max} - \rho\}, D = \frac{f_{\max}}{x_{\min}}$.

**Remark on the competitive ratio $\lambda_{\max}x_{\max}/B$** Although our analysis is against adversarial requests and is worst-case analysis, the competitive ratio result $\lambda_{\max}x_{\max}/B$ still reflects realistic cluster scales, our competitive ratio result capture realistic cluster scales. Consider the following real-world scenario, in a typical multi-tenant hyperscale cluster with $B \approx 60{,}000\text{--}100{,}000$ GPUs, most teams run jobs with at most $x_t \leq 64\text{--}512$ GPUs for durations $\lambda_t \leq 2$ days. If we take one-hour slots (meaning that the minimum duration to rent a GPU is 1 hour), this corresponds to $\bar{\lambda} = 48$ hours and we may set $\lambda_{\max} = 100$ one-hour slots. Under these parameters, $\lambda_{\max}x_{\max}/B$ is well below $1$ (e.g., on the order of $0.06\text{--}0.85$). If one instead uses one-day slots, the factor becomes even smaller.

**Trade-off of the choice of $\lambda_{\max}$.** By employing the alternating block technique with the maximum duration assumption, we obtain a significantly improved competitive ratio. According to Theorem 2, we can set $\lambda_{\max}$ sublinear in $T$. A very small choice of $\lambda_{\max}$ is unhelpful since $\alpha \geq 1$, so it suffices to set $\lambda_{\max}$ such that $\alpha = 1$ in some cases. Importantly, there is an inherent trade-off in selecting $\lambda_{\max}$: a smaller $\lambda_{\max}$ improves the competitive ratio but also increases regret. Setting $\lambda_{\max} = O(T^\gamma)$ for any $0 < \gamma \leq 1$, we get

$$\alpha = \max\left\{1, O\left(\frac{x_{\max}}{B} \cdot T^\gamma\right)\right\}, \quad R(T, P_T) \leq O\left(T^{1-\frac{\gamma}{2}}\right).$$

In practice, we aim to keep the competitive ratio close to $1$ given $x_{\max}$ and $B$, while ensuring that regret remains moderate, for example, by maintaining $\lambda_{\max} \geq O(\sqrt{T})$.

### 4.3 DURATION-ADAPTED DUAL DYNAMIC ALLOCATION ALGORITHM

---

**Algorithm 2** Duration-Adapted Dual Dynamic Allocation Algorithm

---

1: **Input:** Time horizon $T$, total resource $B$, initial dual variable $\nu_1 \geq 0$, and average resource budget $\rho$.
2: **for** $t = 1, \ldots, T$ **do**
3:     Receive request $(f_t, x_t, \lambda_t)$.
4:     Make the primal decision $a_t$:   $\widetilde{a}_t = \arg\max_a a \cdot (f_t(x_t) - \lambda_t \nu_t x_t)$
5:     $\widetilde{B}_t$ and $a_t$ are the same as Line 4 of Algorithm 1
6:     Get the loss function and gradient of the dual variable,

$$w_t(\nu_t) = \nu_t g_t = \begin{cases} \lambda_t \nu_t(\rho - a_t x_t), & \widetilde{B}_t \leq B, \\ 0, & \text{otherwise.} \end{cases}, \quad g_t = \begin{cases} \lambda_t(\rho - a_t x_t), & \widetilde{B}_t \leq B, \\ 0, & \text{otherwise.} \end{cases}$$

7:     Update the dual variable by subroutine online learning algorithm:

$$\nu_{t+1} = \mathrm{OL\text{-}ALG}(\nu_t, w_t(\nu_t))$$

8: **end for**

---

Since $\sum_{\tau=t}^{t+\lambda_t-1} \mu_\tau$ is highly sensitive to $\lambda_t$, the summation may fluctuate significantly as the duration sequence changes. To mitigate this, we average the summation by $\frac{1}{\lambda_t}$ and define the duration-adapted dual variable as $\nu_t = \frac{1}{\lambda_t}\sum_{\tau=t}^{t+\lambda_t-1} \mu_\tau$. Motivated by the Lagrangian formulation in Eq. (5),

we design Algorithm 2. As shown in Algorithm 2, at round $t$, the request is accepted if $\frac{f_t}{\lambda_t x_t} \geq \nu_t$. Here, $\frac{f_t}{\lambda_t x_t}$ represents the revenue per usage volume, while $\nu_t$ serves as the resource price per usage volume, acting as a threshold for deciding whether to accept the request.

**Theorem 3.** *Consider the online reusable resource allocation problem with adversarial requests* $\{f_t, x_t, \lambda_t\}_{t=1}^T$. *If* $\overline{\lambda} = \max_t \lambda_t \leq \lambda_{\max}$ *and* $x_{\max} \leq \frac{B}{3}$, *applying Algorithm 2 initialized with* $\nu_1 \geq 0$ *and* $\rho = B/(3\lambda_{\max})$, *we have for any* $T \geq 1$,

$$\frac{1}{\alpha}\mathrm{OPT}_b(\{a_t^\star\}_{t \in [T]}) - \sum_{t=1}^{T} a_t f_t(x_t) \leq R\big(T, P_T(\{\nu_t^\star\}_{t \in [T]})\big), \tag{8}$$

*where* $\alpha = \max\{1, x_{\max}/\rho\}$, $R\big(T, P_T(\{\nu_t^\star\}_{t \in [T]})\big)$ *is the dynamic regret of the sub-routine* $\mathrm{OL - ALG}$, *satisfying* $\sum_{t=1}^{T} w_t(\nu_t) - \sum_{t=1}^{T} w_t(\nu_t^\star) \leq R\big(T, P_T(\{\nu_t^\star\}_{t \in [T]})\big)$ *and* $P_T(\{\nu_t^\star\}_{t \in [T]}) \leq O\big(\frac{T\overline{\lambda}}{\lambda_{\max}}\big)$.

The proof of Theorem 3 is provided in Appendix A.3. The alternating blocks technique is no longer required when using Algorithm 2. Algorithm 2 achieves an asymptotic competitive ratio of $3\lambda_{\max}x_{\max}/B$ and a bounded path length $P_T(\{\nu_t^\star\}_{t \in [T]}) \leq O\big(\frac{T\overline{\lambda}}{\lambda_{\max}}\big)$, which is nearly as strong as that of the alternating block dual dynamic online allocation algorithm. Moreover, the deterministic regret formulation avoids reliance on expectation-based bounds, which provide only average-case guarantees and may allow poor performance in individual runs. However, this approach requires the additional assumption that the maximum duration $\overline{\lambda}$ is constant; otherwise, the gradient term $\lambda_t(\rho - a_t x_t)$ of the dual variable $\nu_t$ and the path length $P_T$ may become excessively large.

# 5 PRICING EXPERT BASED ALGORITHM

As shown in the previous section, within the primal–dual framework, the dual variables implicitly serve as pricing thresholds, acting as shadow prices that regulate the allocation of limited resources. This observation suggests that a direct pricing approach may also be effective. In this section, we design the *learning from pricing experts* algorithm, which applies a two-sided pricing-threshold method to the online allocation problem.

We model pricing experts $\mathcal{H} = \{h_1, h_2, \ldots, h_H\}$, and each expert corresponds to an interval,

$$h_j = [\phi_j, \phi_{j+1}), \quad \phi_1 = \frac{f_{\min}}{\lambda_{\max}x_{\max}}, \quad \phi_{j+1} = 2\phi_j, \quad \phi_{H+1} = \frac{f_{\max}}{x_{\min}}, \quad j \in [H],$$

which we partition the resource price per usage volume $\frac{f}{\lambda x} \in \left[\frac{f_{\min}}{\lambda_{\max}x_{\max}}, \frac{f_{\max}}{\lambda_{\min}x_{\min}}\right]$ into intervals, and for an expert $j$, it accepts the job request only if the request price per volume falls into this interval. For expert $j$, the acceptance rule is given by

$$\widetilde{a}_t = \begin{cases} 1, & \frac{f_t}{\lambda_t x_t} \in [\phi_j, \phi_{j+1}), \\ 0, & \text{otherwise.} \end{cases} \qquad a_t = \begin{cases} \widetilde{a}_t & \text{if } \widetilde{B}_t \leq B, \\ 0 & \text{otherwise.} \end{cases} \tag{9}$$

We apply the alternating-block technique (Eq. (6)(7)) because each expert needs a fresh start and full replenishment of the budget. At each active block $\mathcal{B}_i$, an expert $h^i \in \mathcal{H}$ is selected according to a distribution $\boldsymbol{p}_i = (p_{i,1}, \ldots, p_{i,H})$, with $\mathbf{Pr}[h^i = h_j] = p_{i,j}$. The reward for block $\mathcal{B}_i$ is defined as the cumulative reward under expert $h^i$, i.e., $u_i(h^i) = \sum_{t \in \mathcal{B}_i} a_t f_t$. We then run a learning with expert advice algorithm to update $\boldsymbol{p}_i$, such as Hedge (Littlestone & Warmuth, 1994; Cesa-Bianchi & Lugosi, 2006; Freund & Schapire, 1996) across active blocks using the rewards $u_i(\cdot)$.

**Theorem 4.** *For the online reusable resource allocation problem with adversarial requests* $\{f_t, x_t, \lambda_t\}_{t=1}^T$, *if* $\overline{\lambda} = \max_t \lambda_t \leq \lambda_{\max}$, *then using the alternating block technique with learning from pricing experts algorithm, we have,*

$$\frac{1}{2\alpha}\mathrm{OPT}_b(\{a_t^\star\}_{t \in [T]}) - \mathbb{E}_{q, j \sim \boldsymbol{p}_i}\left[\sum_{i=1}^{N} u_i(h^j)\right] \leq O\big(\sqrt{T \log H}\big),$$

*where* $\alpha = \frac{(B + (2\lambda_{\max} - 1)x_{\max})H}{B - x_{\max}}$, $H \leq 1 + \left\lceil \log_2\left(\frac{\lambda_{\max}x_{\max}f_{\max}}{\lambda_{\min}x_{\min}f_{\min}}\right)\right\rceil$.

The proof of Theorem 4 is provided in Appendix B.1. Although at first glance the competitive ratio appears worse than that of primal-dual algorithms, this method can be advantageous when $x_{\max}$ is large. In primal–dual methods, $\lambda_{\max}$ must be chosen sublinear in $T$ (and potentially much larger than $\overline{\lambda}$) to ensure sublinear regret, which can degrade the competitive ratio. By contrast, in the expert-based algorithm, we can set $\lambda_{\max} = \overline{\lambda}$ exactly, yielding a stronger guarantee in such cases.

Alternatively, if we set experts to triples as shown in Appendix C.2 Eq. (11). Under this construction, the algorithm achieves an asymptotic competitive ratio of

$$2\alpha = 18 \left(1 + \lceil \log_2(f_{\max}/f_{\min}) \rceil \right) \left(1 + \lceil \log_2(x_{\max}/x_{\min}) \rceil \right) \left(1 + \lceil \log_2(\lambda_{\max}/\lambda_{\min}) \rceil \right).$$

Given information about $\overline{\lambda}$ and the size of the expert set $\mathcal{H}$, one may compare the competitive ratios and regret bounds of these two pricing-expert algorithms with those of primal-dual based algorithms, and select the algorithm that provides the strongest guarantees under the given conditions.

## 6 FLEXIBLE REUSABLE RESOURCE ALLOCATION

We next investigate a flexible, reusable allocation setting, where the platform can decide how much resource to allocate for each job request. Specifically, the platform chooses $x_t \in \{0\} \cup [x_{\min}, x_{\max}]$ for each request $t$, with revenue defined as $f_t(x_t) = f_t \mathbb{I}\{x_t > 0\}$. The assumption $x_t \geq x_{\min}$ when $x \neq 0$ ensures that jobs can be completed within the maximum duration that is still meaningful. Without loss of generality, we scale $x_{\min}, x_{\max}$, and $B$ simultaneously so that $x_{\min} = 1$; the budget constraints remain invariant under such scaling.

This model captures many real-world environments, including LLM inference workloads that allocate varying numbers of GPUs per task, HPC and GPU clusters running moldable batch jobs where more resources reduce runtime (Kalé et al., 2002; Dutton & Mao, 2000), and service systems with adjustable processing effort where higher service rates shorten sojourn times under reusable capacity constraints (Adusumilli & Koole, 2010; Hyon et al., 2020). The allocated resource $x_t$ affects the job duration $\lambda_t(x_t)$, where $\lambda_t(x)$ is adversarially chosen each round but must be non-increasing in $x$ [1]. The platform seeks an allocation sequence $\mathcal{X} = \{x_t\}_{t=1}^T$ that maximizes total revenue:

$$\text{OPT}_a := \max_{\mathcal{X}} \sum_{t=1}^T f_t(x_t), \text{ s.t. } \sum_{\tau=1}^t \mathbb{I}\{\lambda_\tau(x_\tau) \geq t - \tau + 1\} \cdot x_\tau \leq B, \ x_t \in \{0\} \cup [x_{\min}, x_{\max}], \ \forall t$$

Let $\mathcal{X}^\star = \{x_t^\star\}_{t=1}^T$ denote the optimal allocation achieving $\text{OPT}_a$. Compared to the binary allocation case, this model is significantly more challenging: allocating a large resource amount may quickly finish a job but exhausts capacity for future requests, while allocating too little prolongs the job and ties up capacity for many rounds.

**Binary reduction.** We reduce the flexible allocation problem to the binary allocation case by discretizing $[x_{\min}, x_{\max}]$ into powers of two. The possible choices for $x_t$ are drawn from

$$\mathcal{X}_B = \{2^k \mid k \in \mathbb{Z}_{\geq 0}, 2^k \leq B\} \cup \{B\}.$$

Let $x(k)$ denote the resource amount for the $k$-th choice. For $x \in [x_{\min}, x_{\max}]$, we further define

$$\mathcal{X}_m = \{2^k \mid k \in \mathbb{Z}_{\geq 0}, x_{\min} \leq 2^k \leq x_{\max}\} \cup \{x_{\min}, x_{\max}\}, \quad \mathcal{X}_m \subseteq \mathcal{X}_B.$$

At the start, the platform selects $x(k) \in \mathcal{X}_m$ uniformly at random and uses this fixed $x(k)$ throughout; in each round, it either allocates $x_t = x(k)$ or rejects the request.

**Theorem 5.** *Using the binary reduction, let $\text{OPT}_b(x(k), \{a_t^\star\}_{t=1}^T)$ denote the optimal value of the binary case with $x_t = x(k)$. Then,*

$$\frac{\mathbb{E}_{x(k)}[\text{OPT}_b(x(k), \mathcal{A}^\star)]}{\text{OPT}_a(\mathcal{X}^\star)} \geq \frac{1}{2(1 + \lceil \log x_{\max} \rceil)} \geq \frac{1}{2(1 + \lceil \log B \rceil)}.$$

---

[1] If a job is allocated more GPUs, it should complete faster, or at least no slower, than with fewer GPUs.

Theorem 5 allows us to use any of our Algorithms introduced in Sections 4 and 5 for the more complex setting with flexible resource allocation, preserving the guarantees up to a factor of $O(\log B)$. Moreover, we construct a hard instance to show that any *black-box* reduction that fixes a single allocation size $x_t \equiv x_p \in [x_{\min}, x_{\max}]$ incurs a worst-case loss of at least $\frac{2}{1+\lceil \log x_{\max} \rceil}$, since it cannot adapt to heterogeneous jobs and durations. Thus our upper bound is nearly tight—within a constant factor of 4. The proof of Theorem 6 is provided in Appendix C.2.

**Theorem 6** (Lower bound for reduction). *For the flexible allocation problem, suppose the platform uses $x_t = x_p$ for all $t \in [T]$ with $x_p \in [x_{\min}, x_{\max}]$. Then there exists a distribution $\pi$ (independent of $x_p$) and a sufficiently big $T$ such that if the adversary samples a request sequence $\{(f_t, \lambda_t)\}_{t=1}^T$ from $\pi$, we have*

$$\mathbb{E}_\pi \left[ \frac{\text{OPT}_b(x_p, \mathcal{A}^\star)}{\text{OPT}_a(\mathcal{X}^\star)} \right] \leq \frac{2}{1 + \lceil \log x_{\max} \rceil}.$$

## 7 CONCLUSION

We introduced the problem of online reusable resource allocation with adversarial requests and developed algorithms that adapt primal–dual methods, exploit alternating blocks, and incorporate job durations directly into allocation decisions. Complementing these, we proposed a pricing-expert approach that leverages online learning to achieve stronger guarantees in certain regimes. Extending to flexible allocation, we designed a nearly optimal reduction supported by matching lower bounds. These results provide the first systematic framework for adversarial allocation with reusable resources, with direct implications for applications such as GPU and cloud scheduling.

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

## A  PROOFS FOR RESULTS IN SECTION 4

Without loss of generality, we may assume that $f_m \in [0, 1]$ and $x_{\min} = 1$. When there is no ambiguity, we also write $f_m = f_{\max}$, $x_m = x_{\max}$, and $\lambda_m = \lambda_{\max}$ for simplicity.

**Online dual variable learning regret guarantee.**  Let $w_t(\nu_t)$ denote the loss function of the dual variable $\nu_t$. For an online learning algorithm, the regret guarantee is

$$\sum_{t=1}^{T} w_t(\nu_t) - \sum_{t=1}^{T} w_t(\nu) \le R(T, \nu).$$

For a dynamic online learning algorithm, the regret guarantee becomes

$$\sum_{t=1}^{T} w_t(\nu_t) - \sum_{t=1}^{T} w_t(\nu_t^\star) \le R(T, P_T(\nu_1^\star, \dots, \nu_T^\star)),$$

where $P_T(\nu_1, \dots, \nu_T)$ denotes the path-length of the sequence $\nu_1, \dots, \nu_T$.

Online learning algorithms often require bounds on the dual variables $\nu_t$ and the gradients $g_t$. We establish these bounds below. For $t \in [T]$, observe that

$$0 \leq \nu_j \leq \frac{f_m}{x_{\min}}.$$

Indeed, any $\nu_t > f_m/x_{\min}$ cannot affect decisions, since even the maximal revenue per resource unit is less than or equal to $f_m/x_{\min}$; thus we can clip $\nu_t$ to $[0, \ f_m/x_{\min}]$ without loss of generality. Hence,

$$D = \max_{i,j} |\nu_i - \nu_j| \leq \frac{f_m}{x_{\min}}.$$

For the gradient, we have

$$|g_t| = |\rho - a_t x_t| \leq \max\{\rho, \ x_m - \rho\},$$

so $G = \max_{a,x} g = \max_{a,x} \rho - ax \leq \max\{\rho, \ x_m - \rho\}$.

**Lemma 1.** *Let $\mathcal{V} = \{\tau_1, \ldots, \tau_{|\mathcal{V}|}\}$ denote the set of rounds in which the platform's decision violates the budget constraint. Then, for any $\nu > 0$,*

$$\mathrm{OPT}_b\big(\{a_t^\star\}_{t=1}^T\big) - \alpha \sum_{t=1}^T a_t f_t(x_t) \leq |\mathcal{V}| f_{\max} + \alpha \sum_{t=1}^T w_t(\nu) + \alpha R(T, \nu).$$

*Proof.* Let $\mathcal{V} = \{\tau_1, \ldots, \tau_{|\mathcal{V}|}\}$ be the set of rounds in which the decision $\widetilde{a}_t$ violates the budget constraint, i.e.,

$$\forall t \in \mathcal{V}, \quad \sum_{\tau=1}^{t-1} a_\tau \cdot \mathbb{I}\{\lambda_\tau(x_\tau) \geq t - \tau + 1\} \cdot x_\tau + \widetilde{a}_t \cdot x_t > B.$$

For any $t \notin \mathcal{V}$, since

$$a_t = \operatorname*{argmax}_{a \in \{0,1\}} a \cdot \big(f_t(x_t) - \nu_t x_t\big),$$

we obtain

$$f_t(x_t) \geq a_t^\star f_t(x_t) - \nu_t a_t^\star x_t + \nu_t a_t x_t, \qquad a_t f_t(x_t) \geq \nu_t a_t x_t.$$

Define

$$w_t(\nu_t) = \nu_t g_t = \begin{cases} \nu_t(\rho - a_t x_t), & t \notin \mathcal{V}, \\ 0, & t \in \mathcal{V}. \end{cases}$$

Then, for $\alpha \geq 1$

$$\begin{aligned} \alpha a_t f_t(x_t) &= a_t f_t(x_t) + (\alpha - 1) a_t f_t(x_t) \\ &\geq a_t^\star f_t(x_t) - \nu_t a_t^\star x_t + \nu_t a_t x_t + (\alpha - 1) f_t(x_t) \\ &\geq a_t^\star f_t(x_t) - \nu_t a_t^\star x_t + \nu_t a_t x_t + (\alpha - 1) \nu_t a_t x_t \\ &= a_t^\star f_t(x_t) - \alpha \nu_t(\rho - a_t x_t) - \nu_t a_t^\star x_t + \alpha \nu_t \rho \\ &\geq a_t^\star f_t(x_t) - \alpha w_t(\nu_t), \end{aligned} \tag{10}$$

where the last inequality holds by setting $\alpha = \max\left\{1, \frac{x_m}{\rho}\right\}$.

Summing over all $t = 1, \ldots, T$, and using the bounds

$$\sum_{t \in \mathcal{V}} a_t^\star f_t(x_t) \leq |\mathcal{V}| f_{\max}, \ \sum_{t=1}^T w_t(\nu_t) - \sum_{t=1}^T w_t(\nu) \leq R(T, \nu)$$

gives the desired result. $\qquad \square$

## A.1 PROOF OF THEOREM 1

We set $\rho = \frac{B}{T}$. From Lemma 1, we obtain

$$\text{OPT}_b\big(\{a_t^\star\}_{t=1}^T\big) - \alpha \sum_{t=1}^T a_t f_t(x_t) \leq |\mathcal{V}| f_{\max} + \alpha \sum_{t \notin \mathcal{V}} \nu(\rho - a_t x_t) + \alpha R(T, \nu).$$

If $\mathcal{V} = \emptyset$, we simply set $\nu = 0$, which immediately yields the desired bound. Otherwise, if $\mathcal{V} \neq \emptyset$, then there exists some $t \in \mathcal{V}$ such that

$$\sum_{\tau=1}^{t-1} a_\tau \cdot \mathbb{I}\{\lambda_\tau(x_\tau) \geq t - \tau + 1\} \cdot x_\tau + x_t > B.$$

Since

$$\sum_{t \notin \mathcal{V}} a_t x_t \geq \sum_{\tau=1}^{t-1} a_\tau \cdot \mathbb{I}\{\lambda_\tau(x_\tau) \geq t - \tau + 1\} \cdot x_\tau,$$

and noting that $x_t \leq x_m$, we have

$$\sum_{t \notin \mathcal{V}} a_t x_t + x_m > B = T\rho.$$

By choosing $\nu = \frac{f_m}{\alpha \rho}$, it follows that

$$\alpha \sum_{t \notin \mathcal{V}} \nu(\rho - a_t x_t) = \alpha \nu(T - |\mathcal{V}|)\rho - \alpha \nu \sum_{t \notin \mathcal{V}} a_t x_t$$

$$\leq \alpha \nu(T - |\mathcal{V}|)\rho - \alpha \nu(T\rho - x_m)$$

$$= -\alpha \nu |\mathcal{V}|\rho + \alpha \nu x_m$$

$$\leq -|\mathcal{V}| f_m + \alpha f_m.$$

Therefore,

$$\text{OPT}_b\big(\{a_t^\star\}_{t=1}^T\big) - \alpha \sum_{t=1}^T a_t f_t(x_t) \leq \alpha f_m + \alpha R(T, \nu).$$

$\square$

## A.2 PROOF OF THEOREM 2

We partition the entire horizon $T$ into consecutive blocks, each of length equal to the maximum duration $\lambda_m$. Let the odd blocks be collected in the set $\mathcal{T}_{\text{odd}}$, and denote the individual blocks where our algorithm runs as $\mathcal{T}_1, \mathcal{T}_2, \ldots, \mathcal{T}_{N_{\text{odd}}}$. For each block $\mathcal{T}_i$, let $\mathcal{V}_i = \{\tau_1, \ldots, \tau_{|\mathcal{V}_i|}\}$ denote the set of rounds that violate the budget constraint.

Analogous to Lemma 1, for each $i \in [N_{\text{odd}}]$,

$$\text{OPT}_b(\{a_t^\star\}_{t \in \mathcal{T}_i}) - \alpha \sum_{t \in \mathcal{T}_i} a_t f_t(x_t) \leq |\mathcal{V}_i| f_m + \alpha \sum_{t \notin \mathcal{V}_i} \nu_t^\star(\rho - a_t x_t) + \alpha R\big(T, P_T(\{\nu_t^\star\}_{t \in \mathcal{T}_i})\big).$$

If $\mathcal{V}_i = \emptyset$, setting $\nu_t^\star = 0$ for all $t \in \mathcal{T}_i$ gives the bound. Otherwise, when $\mathcal{V}_i \neq \emptyset$, there exists some $t \in \mathcal{V}_i$ such that

$$\sum_{\tau \in \mathcal{T}_i \backslash t} a_\tau \cdot \mathbb{I}\{\lambda_\tau(x_\tau) \geq t - \tau + 1\} \cdot x_\tau + x_t > B.$$

Since

$$\sum_{t \notin \mathcal{V}_i} a_t x_t \geq \sum_{\tau \in \mathcal{T}_i \backslash t} a_\tau \cdot \mathbb{I}\{\lambda_\tau(x_\tau) \geq t - \tau + 1\} \cdot x_\tau,$$

and noting $x_t \leq x_m$, it follows that

$$\sum_{t \notin \mathcal{V}_i} a_t x_t + x_m > B.$$

Let $\rho = B/\lambda_m$, so $B = \rho\lambda_m$. Choosing $\nu_t^\star = \nu_i = \nu = f_m/(\alpha\rho) = f_m/x_m$ for all $t \in \mathcal{T}_i$, we obtain

$$\alpha \sum_{t \notin \mathcal{V}_i} \nu(\rho - a_t x_t) = \alpha\nu(\lambda_m - |\mathcal{V}_i|)\rho - \alpha \sum_{t \notin \mathcal{V}_i} a_t x_t$$
$$\leq \alpha\nu(\lambda_m - |\mathcal{V}_i|)\rho - \alpha\nu(\lambda_m\rho - x_m)$$
$$= -\alpha\nu|\mathcal{V}_i|\rho + \alpha\nu x_m$$
$$\leq -|\mathcal{V}_i|f_m + \alpha f_m.$$

**Bounding the path length.** For the path-length term, note that in vacant blocks the value of $\nu_t$ is inherited from the previous block's value. The number of shifts from 0 to $f_m/(\alpha\rho)$ (or vice versa) is at most $N_{\text{odd}}$, which corresponds to the alternating-shift worst case. Thus,

$$P_T(\nu_1^\star, \ldots, \nu_T^\star) = \sum_{t=2}^T \left| \nu_t^\star - \nu_{t-1}^\star \right| \leq N_{\text{odd}} \cdot \tfrac{f_m}{x_m} \leq \tfrac{\beta f_m}{x_m},$$

where $\beta = \lceil T/\lambda_m \rceil$.

Therefore, for each block,

$$\text{OPT}_b(\{a_t^\star\}_{t \in \mathcal{T}_i}) - \alpha \sum_{t \in \mathcal{T}_i} a_t f_t(x_t) \leq \alpha f_m + \alpha R\big(T, P_T(\{\nu_t^\star\}_{t \in \mathcal{T}_i})\big).$$

Summing over $i = 1, \ldots, N_{\text{odd}}$ yields

$$\text{OPT}_b(\{a_t^\star\}_{t \in \mathcal{T}_{\text{odd}}}) - \alpha \sum_{t \in \mathcal{T}_{\text{odd}}} a_t f_t(x_t) \leq \alpha\beta f_m + \alpha R\big(T, P_T(\{\nu_t^\star\}_{t \in [T]})\big),$$

where $P_T(\{\nu_t^\star\}_{t \in [T]}) \leq \tfrac{\beta f_m}{x_m}$. The same argument applies to the even blocks.

Finally, since

$$\text{OPT}_b(\{a_t^\star\}_{t \in [T]}) \leq \text{OPT}_b(\{a_t^\star\}_{t \in \mathcal{T}_{\text{odd}}}) + \text{OPT}_b(\{a_t^\star\}_{t \in \mathcal{T}_{\text{even}}}),$$

and

$$\mathbb{E}\left[\sum_{t=1}^T a_t f_t(x_t)\right] = \tfrac{1}{2}\left(\sum_{t \in \mathcal{T}_{\text{odd}}} a_t f_t(x_t) + \sum_{t \in \mathcal{T}_{\text{even}}} a_t f_t(x_t)\right),$$

we conclude that

$$\text{OPT}_b(\{a_t^\star\}_{t \in [T]}) - 2\alpha \cdot \mathbb{E}\left[\sum_{t=1}^T a_t f_t(x_t)\right] \leq \alpha\beta f_m + \alpha R\big(T, P_T(\{\nu_t^\star\}_{t \in [T]})\big),$$

with $P_T(\{\nu_t^\star\}_{t \in [T]}) \leq \tfrac{\beta f_m}{x_m}$.

If we set $\rho = \tfrac{B}{2\lambda_m}$, and make the assumption that $x_m \leq \tfrac{B}{2}$, then $B = 2\rho\lambda_m$ by choosing $\nu_t^\star = \nu_i = \nu = \tfrac{f_m}{\alpha\rho}, t \in \mathcal{T}_i$, we have

$$\alpha \sum_{t \notin \mathcal{V}_i} \nu(\rho - a_t x_t) = \alpha\nu(\lambda_m - |\mathcal{V}_i|)\rho - \alpha \sum_{t \notin \mathcal{V}_i} a_t x_t$$
$$\leq \alpha\nu(\lambda_m - |\mathcal{V}_i|)\rho - \alpha\nu(2\lambda_m\rho - x_m)$$
$$= -\alpha\nu|\mathcal{V}_i|\rho - \alpha\nu\frac{B}{2} + \alpha\nu x_m$$
$$\leq -|\mathcal{V}_i|f_m.$$

Analogously, we get

$$\text{OPT}_b(\{a_t^\star\}_{t \in [T]}) - 2\alpha \cdot \mathbb{E}\left[\sum_{t=1}^T a_t f_t(x_t)\right] \leq \alpha R\big(T, P_T(\{\nu_t^\star\}_{t \in \mathcal{T}_i})\big),$$

where $P_T(\{\nu_t^\star\}_{t \in [T]}) \leq \tfrac{\beta f_m}{x_m}$.

Under the additional assumption $x_m \leq B/2$, setting $\rho = B/(2\lambda_m)$ and $\nu = f_m/(\alpha\rho)$ strengthens the bound and removes the additive $\alpha f_m$ term.

**Intuition of alternating block** The proof relies on partitioning the horizon into odd and even blocks, each of length $\lambda_m$. By running the algorithm only on odd (or only on even) blocks, we ensure that resource usage from one block does not interfere with the next, since any accepted request must finish within $\lambda_m$ rounds. The factor of 2 in the final bound arises because the algorithm is randomized between odd and even blocks, and hence its expected reward is the average of the two cases.

### A.3 Proof for Theorem 3

*Proof.* Let $\mathcal{V} = \{\tau_1, \ldots, \tau_{|\mathcal{V}|}\}$ denote the set of rounds where the tentative action $\widetilde{a}_t$ violates the budget constraint, i.e.,

$$\forall t \in \mathcal{V}, \quad \sum_{\tau=1}^{t-1} a_\tau \cdot \mathbb{I}\{\lambda_\tau(x_\tau) \geq t - \tau + 1\} \cdot x_\tau + \widetilde{a}_t \cdot x_t > B.$$

We denote the auxiliary dual variables as $\mu_1, \mu_2, \ldots, \mu_T$, and define

$$\nu_t = \frac{1}{\lambda_t} \sum_{\tau=t}^{t+\lambda_t-1} \mu_\tau.$$

The loss function of the dual variable $\nu_t$ is given by

$$w_t(\nu_t) = \nu_t g_t = \begin{cases} \lambda_t \nu_t (\rho - a_t x_t), & t \notin \mathcal{V}, \\ 0, & t \in \mathcal{V}. \end{cases}$$

Applying Lemma 1, we obtain

$$\mathrm{OPT}_b(\{a_t^\star\}_{t\in[T]}) - \alpha \sum_{t=1}^{T} a_t f_t(x_t) \leq |\mathcal{V}| f_m + \alpha \sum_{t\notin\mathcal{V}} \lambda_t \nu_t^\star(\rho - a_t x_t) + \alpha R(T, P_T(\{\nu_t^\star\}_{t\in[T]})).$$

**Step 1: Decomposition of the dual loss term.**

$$\sum_{t\notin\mathcal{V}} \lambda_t \nu_t(\rho - a_t x_t) = \sum_{t\notin\mathcal{V}} \sum_{\tau\in[l(t),t],\,\tau\notin\mathcal{V}} (\rho - \mathbb{I}[\lambda_\tau \geq t-\tau+1]a_\tau x_\tau)\mu_t + \sum_{t\in\mathcal{V}} \sum_{\tau\in[l(t),t-1],\,\tau\notin\mathcal{V}} (\rho - \mathbb{I}[\lambda_\tau \geq t-\tau+1]a_\tau x_\tau)\mu_t,$$

where $l(t) = \mathrm{argmin}_l\{\lambda_l \geq t - l + 1\}$.

For $t \notin \mathcal{V}$, we set $\mu_t = 0$. For $t \in \mathcal{V}$, since $\widetilde{a}_t$ violates the budget constraint, we have

$$\sum_{\tau\in[l(t),t-1],\,\tau\notin\mathcal{V}} \mathbb{I}[\lambda_\tau \geq t-\tau+1]a_\tau x_\tau + x_t > B.$$

**Step 2: Choice of parameters.** Since $t - 1 - l(t) + 1 \leq \lambda_m$, with $\rho = \frac{B}{3\lambda_m}$, and $x_m \leq B/3$, we choose

$$\mu_t = \frac{f_m}{\alpha\lambda_m\rho} = \frac{f_m}{\lambda_m x_m}.$$

Then

$$f_m + \alpha \sum_{\tau\in[l(t),t-1],\,\tau\notin\mathcal{V}} (\rho - \mathbb{I}[\lambda_\tau \geq t-\tau+1]a_\tau x_\tau)\mu_t$$

$$\leq f_m + \alpha\lambda_m\mu_t\rho - \alpha(B - x_m)\mu_t$$

$$= \left(f_m - \tfrac{1}{3}\alpha B\mu_t\right) + \left(\alpha\lambda_m\mu_t\rho - \tfrac{1}{3}\alpha B\mu_t\right) + \left(\alpha x_m\mu_t - \tfrac{1}{3}\alpha B\mu_t\right)$$

$$\leq \left(f_m - \alpha\lambda_m\mu_t\rho\right) + 0 + 0$$

$$\leq 0.$$

**Step 3: Final bound.**

$$\text{OPT}_b(\{a_t^\star\}_{t\in[T]}) - \alpha \sum_{t=1}^T a_t f_t(x_t) \le \sum_{t\in\mathcal{V}} f_m + \alpha \sum_{\tau\in[l(t),t-1],\,\tau\notin\mathcal{V}} (\rho - \mathbb{I}[\lambda_\tau \ge t - \tau + 1]a_\tau x_\tau)\mu_t + \alpha R(T, P_T(\{\nu_t^\star\}_{t\in[T]}))$$

$$\le \alpha R(T, P_T(\{\nu_t^\star\}_{t\in[T]})).$$

**Step 4: Bounding the path length.** Since $\lambda_m \ge \max_t \lambda_t$ is a tunable hyperparameter, and letting $\overline{\lambda} = \max_t \lambda_t$ denote the actual maximum duration (a constant), we have

$$P_T(\nu_1^\star,\ldots,\nu_T^\star) = \sum_{t=2}^T \left|\nu_t^\star - \nu_{t-1}^\star\right| = \sum_{t=2}^T \left|\sum_{\tau=t}^{t+\lambda_t-1}\mu_\tau - \sum_{\tau=t+1}^{t+\lambda_{t+1}}\mu_\tau\right| < \sum_{t=2}^T \overline{\lambda}\cdot\frac{f_m}{\lambda_m x_m} = O\left(\frac{T\overline{\lambda}}{\lambda_m}\right).$$

Thus,

$$R(T, P_T(\{\nu_t^\star\}_{t\in[T]})) \le O\left(\sqrt{T(1 + P_T)}\right).$$

Finally, setting $\lambda_m = O(T^\gamma)$ for any $0 < \gamma \le 1$, and applying subroutine *Ader* algorithm in Zhang et al. (2018a), we obtain

$$R(T, P_T) \le O\left(T^{1-\frac{\gamma}{2}}\right).$$

$\square$

# B PROOFS FOR RESULTS IN SECTION 5

## B.1 PROOF FOR THEOREM 4

*Proof.* Denote the set of all active blocks by

$$\mathcal{B} = \mathcal{B}_1 \cup \mathcal{B}_2 \cup \cdots \cup \mathcal{B}_N.$$

We discretize the pricing thresholds into experts. Define threshold intervals

$$h_j = [\phi_j, \phi_{j+1}),\quad \phi_1 = \frac{f_{\min}}{\lambda_m x_m},\quad \phi_{j+1} = 2\phi_j,\quad \phi_{H+1} = \frac{f_m}{\lambda_{\min} x_{\min}},\quad h_H = [\phi_H, \phi_{H+1}],$$

and let $\mathcal{H} = \{h_1, h_2, \ldots, h_{|\mathcal{H}|}\}$.

Under expert $h_j$, the decision maker accepts a job whenever its per usage volume revenue satisfies

$$\widetilde{a}_t = \begin{cases} 1, & \frac{f_t}{\lambda_t x_t} \in [\phi_j, \phi_{j+1}), \\ 0, & \text{otherwise.} \end{cases}$$

**Step 1: Performance of a single expert.** Let $\mathcal{T}_j = \{t : \frac{f_t}{\lambda_t x_t} \in [\phi_j, \phi_{j+1})\}$. Define the violation set $\mathcal{V}_j \subseteq \mathcal{T}_j$, and for each $t \in \mathcal{T}_j$ let

$$B_t = \sum_{\tau\in\mathcal{T}_j,\tau\le t} a_\tau \cdot \mathbb{I}\{\lambda_\tau(x_\tau) \ge t - \tau + 1\} \cdot x_\tau, \qquad r_t = B_t\phi_j.$$

If $t \in \mathcal{V}_j$, then $B_t + x_t > B$. Since $\frac{f_t}{\lambda_t x_t} \in [\phi_j, \phi_{j+1})$, we have

$$r_t = B_t\phi_j \ge \tfrac{1}{2}(B - x_m)\phi_{j+1}.$$

Moreover,

$$\sum_{t\in\mathcal{T}_j} a_t f_t = \sum_{t\in\mathcal{T}_j} a_t \frac{f_t}{\lambda_t x_t}\lambda_t x_t \ge \phi_j \sum_{t\in\mathcal{T}_j} a_t \lambda_t x_t = \sum_{t\in\mathcal{T}_j} r_t \ge \sum_{t\in\mathcal{V}_j} r_t.$$

Since $f_t \leq \lambda_m x_m \phi_{j+1}$, we conclude that

$$\sum_{t \in \mathcal{V}_j} f_t \leq \frac{2\lambda_m x_m}{B - x_m} \sum_{t \in \mathcal{V}_j} r_t \leq \frac{2\lambda_m x_m}{B - x_m} \sum_{t \in \mathcal{T}_j} a_t f_t.$$

Thus,

$$\sum_{t \in \mathcal{T}_j} a_t f_t \geq \text{OPT}_b(\{t \in \mathcal{T}_j\}) - \sum_{t \in \mathcal{V}_j} f_t \geq \text{OPT}_b(\{t \in \mathcal{T}_j\}) - \frac{2\lambda_m x_m}{B - x_m} \sum_{t \in \mathcal{T}_j} a_t f_t.$$

Rearranging,

$$\sum_{t \in \mathcal{T}_j} a_t f_t \geq \frac{B - x_m}{B + (2\lambda_m - 1)x_m} \cdot \text{OPT}_b(\{t \in \mathcal{T}_j\}).$$

**Step 2: Selecting the best expert.**    Since

$$\sum_{j=1}^{|\mathcal{H}|} \text{OPT}_b(\{t \in \mathcal{T}_j\}) \geq \text{OPT}_b(\{a_t^\star\}_{t \in \mathcal{B}}),$$

by the pigeonhole principle there exists an expert $j^\star$ such that

$$\text{OPT}_b(\{t \in \mathcal{T}_{j^\star}\}) \geq \frac{\text{OPT}_b(\{a_t^\star\}_{t \in \mathcal{B}})}{|\mathcal{H}|}.$$

And

$$|\mathcal{H}| \leq 1 + \left\lceil \log_2 \left( \frac{\lambda_m x_m f_m}{\lambda_{\min} x_{\min} f_{\min}} \right) \right\rceil$$

**Step 3: Hedge analysis.**    At each block $i$, the algorithm selects an expert $j_i$. Denote the cumulative reward obtained using expert $j_i$ in $\mathcal{B}_u$ as $u_i(j_i)$. By the regret guarantee of Hedge (or any near-optimal expert algorithm), we have

$$\sum_{i=1}^{N} u_i(j^\star) - \mathbb{E}_{j_i \sim p_i}\left[ \sum_{i=1}^{N} u_i(j_i) \right] \leq O\left( \sqrt{T \log |\mathcal{H}|} \right).$$

Since $\sum_{t \in \mathcal{T}_{j^\star}} a_t f_t = \sum_{i=1}^{N} u_i(j^\star)$, and $\sum_{t \in \mathcal{B}} a_t f_t = \sum_{i=1}^{N} u_i(j_i)$, it follows that

$$\frac{1}{\alpha} \text{OPT}_b(\{a_t^\star\}_{t \in \mathcal{B}}) - \mathbb{E}_{j_i \sim p_i}\left[ \sum_{t \in \mathcal{B}} a_t f_t \right] \leq O\left( \sqrt{T \log |\mathcal{H}|} \right),$$

where

$$\alpha = \frac{(B + (2\lambda_m - 1)x_m) H}{B - x_m}.$$

**Step 4: Alternating block scheme.**    Finally, due to the alternating block scheme (analogous to Proof A.2), we obtain

$$\frac{1}{2\alpha} \text{OPT}_b(\{a_t^\star\}_{t \in [T]}) - \mathbb{E}_{j_i \sim p_i}\left[ \sum_{i=1}^{N} u_i(j_i) \right] \leq O\left( \sqrt{T \log |\mathcal{H}|} \right),$$

where $\alpha = \frac{(B + (2\lambda_m - 1)x_m)H}{B - x_m}$, $H = |\mathcal{H}| \leq 1 + \left\lceil \log_2 \left( \frac{\lambda_m x_m f_m}{\lambda_{\min} x_{\min} f_{\min}} \right) \right\rceil$.

$\square$

**Alternative experts**  If we set the expert as

$$h_{j,k,l} := \{[F_j, F_{j+1}), [X_k, X_{k+1}), [L_l, L_{l+1})|j, k, l\} \tag{11}$$

where

$$\phi_1 = f_{\min}, \quad , \phi_{j+1} = 2\phi_j, \quad X_1 = x_{\min}, \quad X_{k+1} = 2X_k, \quad L_1 = \lambda_{\min} \geq 2, \quad L_{k+1} = 2L_k,$$

and let

$$\mathcal{H} = \{h_{j,k,l}|j \leq 1 + \lceil\log_2 \frac{f_m}{f_{\min}}\rceil, k \leq 1 + \lceil\log_2 \frac{x_m}{x_{\min}}\rceil, l \leq 1 + \lceil\log_2 \frac{\lambda_m}{\lambda_{\min}}\rceil\}.$$

Under expert $h_{j,k,l}$, the platform accepts a job whenever the request $(f_t, x_t, \lambda_t)$ satisfies

$$\widetilde{a}_t = \begin{cases} 1, & f_t \in [F_j, F_{j+1}), \text{and } x_t \in [X_k, X_{k+1}), \text{and } \lambda_t \in [L_j, L_{j+1}) \\ 0, & \text{otherwise.} \end{cases}$$

Analogous to the above proof, we can get that

$$\frac{1}{2\alpha} \cdot \text{OPT}_b(\{a_t^\star\}_{t\in[T]}) - \mathbb{E}_{q,j_i\sim p_i}\left[\sum_{t=1}^T a_t f_t(x_t)\right] \leq O\left(\sqrt{T \log |\mathcal{H}|}\right).$$

where $\alpha = 9H$ and

$$H = |\mathcal{H}| \leq \left(1 + \left\lceil\log_2 \frac{f_m}{f_{\min}}\right\rceil\right)\left(1 + \left\lceil\log_2 \frac{x_m}{x_{\min}}\right\rceil\right)\left(1 + \left\lceil\log_2 \frac{\lambda_m}{\lambda_{\min}}\right\rceil\right).$$

## C  PROOFS FOR RESULTS IN SECTION 6

### C.1  PROOF FOR THEOREM 5

*Proof.* Given the optimal solution $\mathcal{X}^\star$ and $\text{OPT}_a(\mathcal{X}^\star)$, we define

$$T_k = \{t \mid x_t^\star \in (x(k-1), x(k)]\} = \{\tau_{k,1}, \cdots, \tau_{k,|T_k|}\}$$

where $x(-1) = 0, x(K+1) = x_m$. For a specific $k$, we ignore the subscript $k$.

Let $\text{OPT}_b(\mathcal{A}^\star, x(k), T_k)$ be the optimal value only do the optimization in $T_k$ with budget constraints. For $t \in T_k$, we round $x_t^\star \in (x(k-1), x(k)]$ to $x(k)$. We partition $T_k$ into $T_{k,1}$ and $T_{k,2}$ in the following way:

1. $T_{k,1} = \emptyset, T_{k,2} = \emptyset$

2. For $j = 1, \cdots, |T_k|$,

    (a) If $T_{k,1} \cup \tau_{k,j}$ does not violate the constraints, the cons $\sum_{\ell \in T_{k,1} \cup \tau_{k,j}} \mathbb{I}\{\lambda_\ell(x(k)) \geq t - \ell + 1\} \cdot x(k) \leq B, t \leq \tau_{k,j}$, then $T_{k,1} = T_{k,1} \cup \tau_{k,j}$; Otherwise $T_{k,2} = T_{k,2} \cup \tau_{k,j}$.

3. We get $T_{k,1}$ and $T_{k,2}$.

We prove that $T_{k,1}$ does not violate budget constraints and $T_{k,2}$ does not violate budget constraints. If not, then during the above process, there exists $j \in [|T_k|]$, that for some $t$,

$$\sum_{\ell \in T_{k,1} \cup \tau_{k,j}} \mathbb{I}\{\lambda_\ell(x(k)) \geq t - \ell + 1\} \cdot x(k) > B, \qquad \sum_{\ell \in T_{k,2} \cup \tau_{k,j}} \mathbb{I}\{\lambda_\ell(x(k)) \geq t - \ell + 1\} \cdot x(k) > B.$$

Adding these two inequality together, we get

$$\sum_{\ell \in \{\tau_{k,1}, \cdots, \tau_{k,j}\}} \mathbb{I}\{\lambda_\ell(x(k)) \geq t - \ell + 1\} \cdot x(k) > 2B.$$

We know that for $t \in T_k$, $x_t^\star > x(k-1) = \frac{1}{2}x(k)$, and and $\lambda_t(x)$ is non-increasing with $x$ increase, hence

$$\sum_{\ell \in \{\tau_{k,1}, \cdots, \tau_{k,j}\}} \mathbb{I}\{\lambda_\ell(x_\ell^\star) \geq t - \ell + 1\} \cdot x_\ell^\star > B,$$

which is contradict to the optimal solution does not violate budget constraint.

For $t \in T_{k,1}$, we round $x_t^\star \in (x(k-1), x(k)]$ to $x(k)$, and let

$$a_t = \begin{cases} 1, & t \in T_{k,1} \\ 0, & t \notin T_{k,1} \end{cases}$$

This $\{a_t\}_{t=1}^T$ is a feasible solution according to our seperation of $T_{k,1}$ and $T_{k,2}$. Therefore

$$\mathrm{OPT}_b\left(\mathcal{A}^\star, x(k)\right) \geq \mathrm{OPT}_b\left(\mathcal{A}^\star, x(k), T_k\right) \geq \mathrm{OPT}_a\left(\mathcal{X}^\star, T_{k,1}\right).$$

Analogously,

$$\mathrm{OPT}_b\left(\mathcal{A}^\star, x(k)\right) \geq \mathrm{OPT}_a\left(\mathcal{X}^\star, T_{k,2}\right).$$

So,

$$2\sum_{k=1}^{K+1} \mathrm{OPT}_b\left(\mathcal{A}^\star, x(k)\right) \geq \sum_{k=1}^{K+1} \mathrm{OPT}_a\left(\mathcal{X}^\star, T_{k,1}\right) + \mathrm{OPT}_a\left(\mathcal{X}^\star, T_{k,2}\right) = \mathrm{OPT}_a\left(\mathcal{X}^\star\right).$$

Hence,

$$\mathbb{E}_{x(k)}\left[\mathrm{OPT}_b\left(\mathcal{A}^\star, x(k)\right)\right] \geq \frac{1}{2(K+1)}\mathrm{OPT}_a\left(\mathcal{X}^\star\right) \geq \frac{1}{2\left(1 + \lceil\log x_m\rceil\right)}\mathrm{OPT}_a\left(\mathcal{X}^\star\right).$$

$\square$

### C.2 PROOF OF THEOREM 6

*Proof.* We construct an adversarial environment instance to establish the lower bound. Suppose $x_{\min} \geq 1$, and $x_m = 2^K$ or $K$ satisfies $2^K < x_m < 2^{K+1}$. The time horizon is $T$, divided into $N$ batches such that $T = NT_1$ rounds, with $T_1 = \lfloor B \rfloor + T_0$ and $T_0 \gg B$.

With a slight abuse of notation, define

$$f_k = 2^k, \quad x_k = x(k) = 2^k \text{ for } k \leq K, \quad x_{K+1} = x(K+1) = x_m, \quad \lambda_k(x) = \begin{cases} T, & 0 < x < x_k, \\ T_0, & x \geq x_k. \end{cases}$$

Within block $i$, let $t_i$ denote the local time index (the global index is $t = (i-1)T_1 + t_i$). At the beginning of the process, the adversary specifies the instance distribution $\pi$ by drawing $k_i \sim \mathrm{Unif}([K+1])$ independently for each block $i$. In block $i$, the adversary generates valid requests of size $x(k_i)$ and fills the remaining rounds with null jobs, i.e., requests that consume zero resource and yield zero reward.

$$M_k = \left\lfloor \frac{B}{2^k} \right\rfloor, \qquad f_{t_i} = \begin{cases} f_k, & t_i \in [M_k], \\ 0, & t_i \in [M_k + 1, T_1], \end{cases} \qquad \lambda_{t_i} = \begin{cases} \lambda_k, & t_i \in [M_k], \\ \text{null}, & t_i \in [M_k + 1, T_1]. \end{cases}$$

The optimal flexible solution achieves all possible revenue by setting $x_{t_i} = x_{k_i}$ at each block and accepting all valid requests. Since the budget is fully replenished at the end of each block, the optimal cumulative revenue per block is at least $2^K$.

Now consider the reduction strategy where the platform fixes $x_p \in [x_{k_p}, x_{k_p+1})$. If it accepts any request with $x_{t_i} \geq x_{k_p+1}$, it consumes $x_p$ units of resource unnecessarily, which leads to a loss of at least $\frac{1}{K+1} \cdot 2^{k_p}$ in expectation in each subsequent block—because it will accept at least one fewer job when $k_i = k_p$. Over $(N-i)$ future blocks, this yields a cumulative loss of $\frac{N-i}{K+1} \cdot 2^{k_p}$. For sufficiently large $N$, such a choice is suboptimal. Hence, under the reduction strategy $x_t = x_p$, the platform will only accept requests with $x_{t_i} \leq x_{k_p}$.

Within each block, the platform can complete at most $\lfloor B/x_{k_p} \rfloor = 2^{K-k_p}$ jobs due to the budget constraint. Therefore, the expected cumulative revenue is bounded by

$$\frac{1}{K+1} \cdot 2^{K-k_p} \cdot \sum_{j=0}^{k_p} 2^j < \frac{1}{K+1} \cdot 2^{K-k_p} \cdot 2^{k_p+1} = \frac{2}{K+1} \cdot 2^K.$$

Since the optimal flexible solution collects at least $2^K$ revenue per block, we obtain

$$\mathbb{E}_\pi \left[ \frac{\mathrm{OPT}_b(x_p, \mathcal{A}^\star)}{\mathrm{OPT}_a(\mathcal{X}^\star)} \right] \leq \frac{2}{K+1} = \frac{2}{1 + \lceil \log x_m \rceil}.$$

$\square$

## USE OF LARGE LANGUAGE MODELS (LLMS)

We used large language models (LLMs) solely as a writing-assistance tool. In particular, LLMs were employed to polish the grammar, clarity, and style of the manuscript. All research ideas, modeling assumptions, algorithmic designs, theoretical results, and experiments were conceived, developed, and validated entirely by the authors. The role of the LLM was limited to improving the readability of the text and did not contribute to the research content.

