# OpenReview forum: "Online Reusable Resource Allocation with Adversarial Requests"
_ICLR.cc/2026/Conference — Submitted to ICLR 2026_

### Official Review · Reviewer_1QE6 · 2025-10-28

**Soundness:** 3
**Presentation:** 3
**Contribution:** 3
**Rating:** 6
**Confidence:** 3

**Summary:**

The paper considers an interval selection problem. The way I see the problem. We are given a rectangular grid. Time goes from left to right. At each integer time t, we see an interval in row $x_t$, starting at t, and with some profit. Rows correspond to resources. If that interval does not overlap a previously accepted interval in this row, the we can decide to accept it or not. Accepted intervals are active as long as their right end is not reached. The goal is to maximize the total profit of accepted intervals subject to the total accepted active interval length not exceeding given budget B at any time.

The performance of an algorithm is measured by the competitive ratio, which is the solution one could compute if all future arriving intervals were known in advance.

Each row is a problem by its own, but the budget B is shared among them.

Two classes of algorithms are studied. The first one uses the primal-dual framework, with Lagrangian relaxation, and sub-gradient descent. The second the pricing expert framework. Clearly for the problem, the density of an interval, profit over interval length, is of importance, and simple reasonable algorithms accept whenever the density is above some threshold. This threshold might vary over time.

In the primal dual framework, a promise is considered that the interval never exceed some $\lambda_\max$ in length. The time line is then divided into blocks of size $\lambda_\max$ each. The algorithm makes decisions only in even or odd indexed blocks according to an initially chosen random bit. In that sense it solves an independent online knapsack problem in each of those blocks.

In the expert framework, each expert accepts intervals whose density is with some value interval. Every expert has its own value interval. The multiplicative weight update method is used to learn the best expert.

Finally a version is considered where the algorithm can decide on the amount of resource to allocate to each accepted interval, more amount reduces the interval length.

**Strengths:**

I like the problem a lot, once I understood correctly. I like that two approaches are studied and a variant.

**Weaknesses:**

I think some discussion is missing with other online packing problems, and similarities in the algorithms. But this would mean to remove some of the interesting technical part of the paper.

**Questions:**

Page 2 line 068. I don't understand how the competitive ratio can depend on $x_\max$, which is the largest index of the resource in the instance. The problem should be invariant to numbering the resources. Hence it might be that I miss-understood something in the problem description. After reading Section 3 I think you mean that there is a single resource available with quantity B. Every interval comes with a resource requirement $x_t$ which is a quantity, not a resource identifier. The paper could do a better job in explaining the problem.

I am wondering if the restriction of having a single request at every time step makes the problem easier?

Notation. Why do you need $f_t, \lambda_t$ to be functions of $x_t$. They could just be values.
Page 5 line 240. I find this a complicated way to write $\tilde a_t=1$ iff $f_t \geq \mu_t x_t$.

---

> ### Author Response · Authors · 2025-11-21
>
> >I think some discussion is missing with other online packing problems, and similarities in the algorithms. But this would mean to remove some of the interesting technical part of the paper.
>
>
> We thank the reviewer for this helpful suggestion. We will add a concise discussion of the classical literature on online packing. In particular, classical online packing models typically focus on non-reusable resources and binary accept/reject decisions, which can be viewed as a special case of the online non-reusable allocation setting, and we already discuss some of these works in our related-work section.
>
> In the revision (where ICLR allows one additional page during the discussion/rebuttal phase), we will add a paragraph in the related work section. Concretely, we plan to add one or more paragraphs covering at least the following:
>
>
> > A large body of work on online packing and online linear programming focuses on the non-reusable resource setting. This line of research includes online matching and AdWords-type models, where advertisers have fixed budgets and impressions arrive online; primal–dual algorithms with threshold prices achieve near-optimal guarantees in these settings (Mehta et al., 2007; Buchbinder et al., 2007). In the classical model of online bipartite matching with adversarial arrival, Karp et al. (1990) showed that a simple greedy algorithm achieves a $1/2$-competitive ratio, and that this can be improved to $1 - 1/e$ in the random-order model using the Ranking algorithm. Building on this, the AdWords problem generalizes online matching to incorporate heterogeneous budgets and multi-keyword bids while still treating budgets as non-replenishing resources; Mehta et al. (2007) and Buchbinder et al. (2007) design primal–dual online algorithms that achieve the optimal $1 - 1/e$ competitive ratio in appropriate small-bid regimes. More generally, online packing LP formulations and their stochastic or random-order variants obtain $(1 - o(1))$-competitive ratios by repeatedly solving or approximately updating dual prices, often via online learning or mirror-descent style updates (Feldman et al., 2009; Agrawal et al., 2014; Devanur et al., 2019; Gupta & Molinaro, 2016; Balseiro et al., 2023). A related stream in revenue management and online knapsack studies LP-based resolving heuristics against a deterministic LP)benchmark: Jasin and Kumar (2012) show that frequently re-solving the LP yields constant regret under a non-degeneracy condition, while Bumpensanti and Wang (2020) design an infrequent re-solving policy with uniformly bounded loss without requiring non-degeneracy. Arlotto and Gurvich (2019) obtain uniformly bounded regret for the multisecretary problem, another canonical online packing model. More recent work develops general constant-regret frameworks for stochastic online allocation and packing—Vera and Banerjee (2020), Vera et al. (2021), and Banerjee and Freund (2020) use Bellman inequality or uniform loss’ arguments to design policies that repeatedly solve LP or DP relaxations and achieve regret that does not grow with the horizon or capacities.
>
> We will also include the corresponding references in the bibliography in the revised version.
>
> > Every interval comes with a resource requirement $x_t$ which is a quantity, not a resource identifier. The paper could do a better job in explaining the problem.
>
> We thank the reviewer for pointing out this source of confusion. In our formulation, there is a single resource type, and $x_t$ denotes the amount of this resource required by job request $t$, rather than an identifier of a particular resource. In the revision, we will clarify this early in the model section by explicitly stating that we consider a single homogeneous resource and that $x_t$ represents its required amount for each job request.

---

> ### Author Response · Authors · 2025-11-21
>
> > I am wondering if the restriction of having a single request at every time step makes the problem easier?
>
> We thank the reviewer for this question. The single arrival per time step setting is standard in the online allocation literature, and we adopt it primarily to keep the model clean and convey the main ideas. Even under this restriction, the adversarial reusable-resource setting we study remains technically challenging: decisions are irrevocable, durations create long-range coupling across time.
>
> Allowing multiple requests to arrive simultaneously (e.g., an arbitrary number of requests per time step in an adversarial fashion) is, at least superficially more general and harder. In many cases, such batched arrivals can be reduced to the single-arrival model by processing requests in an arbitrary order within each time step, but formally analyzing the most general multi-request adversarial model would require additional technical work. We view this extension as a natural and interesting direction for future research, and will add a short remark to the paper to clarify this point.
>
> > Notation. Why do you need $f_t$ and $\lambda_t$ to be functions of $x_t$? They could simply be values. Page 5, line 240: this seems like an unnecessarily complicated way to express $\tilde{a}_t = 1$ iff $\tilde{a}_t = 1$ iff $f_t \ge \mu_t x_t$.
>
> You are right that in Sections 4 and 5 it is not necessary to treat $f_t$ and $\lambda_t$ as functions of $x_t$; they can indeed be viewed as scalar values there. Our intent was to keep the notation consistent with Section 6, where the dependence on $x_t$ becomes important. In the revision, we will simplify the notation in Sections 4 and 5 by treating $f_t$ and $\lambda_t$ as values, and only introduce the functional dependence when it is actually needed in Section 6.
>
> Regarding line 240, we agree that the condition could be written more simply as you suggested. We originally chose the more elaborate expression to maintain consistent with the Lagrangian formulation.

---

### Official Review · Reviewer_tJEB · 2025-11-01

**Soundness:** 3
**Presentation:** 2
**Contribution:** 2
**Rating:** 4
**Confidence:** 3

**Summary:**

The paper studies the online resource allocation problem, where the learner receives a sequence of job requests (gain, duration, resource cost) and must determine immediately whether to accept or reject each request. The paper proposes two different approaches, one based on Lagrange duality and the other based on prediction with expert advice. Both approaches are shown, via mathematical analyses, to be provably effective and computationally efficient.

**Strengths:**

The paper studies an interesting problem and proposes effective solutions for it. The theoretical results on competitive ratio seem to be optimal. The proof strategies also appear to be novel and could be of independent interest. I verified the correctness of the proofs of Theorem 1 and Theorem 2.

**Weaknesses:**

Please address the following concerns and questions:

*Theorem 1:*
- Applying regret bound on OLG-ALG without specifying the set of $\nu$: assume $\nu$ belongs to a set $V$. Online learning algorithms generally require $V$ to be bounded and convex. However, it seems that this set $V$ is not defined anywhere.
- Lacking a quantifier for $R(T, \nu)$ everywhere: in many places, the regret bound $R(T, \nu)$ is used without any quantifier and undefined $\nu$. Can *any* $\nu$ be used here, or should it be $\max_{\nu \in V}R(T, \nu)$? I suspect it should be $\max_{\nu \in V}R(T, \nu)$ everywhere, because you are plugging a specific value of $\nu$ in your bound (line 685).
- In line 65, the paper claimed that they adaptively learn and use a sequence of $(\nu_t)_t$, which is contrary to the non-reusable setting that uses a single $\nu^\star$. However, if my understanding of the proofs in the appendix is correct, then it is in fact the case that the regret bounds are used with respect to a single $\nu$. For example, in Algorithm 1, the regret bound is a static bound that holds for a single $\nu$.
- Line 269, at the end of page 5: I believe writing $R(T, \nu) \leq O(\sqrt{T})$ is incomplete. There should be a factor on the scale of the losses here.

*Theorem 2*:
- Continue with the point above, in the proof of Theorem 2, despite the theorem using a path-length bound,  all the comparators in active blocks $\nu^\star_t$ are  set to be equal to $f_m / x_m$ (in line 721). Shouldn't this defeat the purpose of using a dynamic regret bound?
- Line 729 to 734: how is this bound on $P_T$ correct? If $\nu^\star_t = f_m / x_m$ and $\nu^\star_{t+1} = \nu^*_t$ for all $t \in \mathcal{T}_i$ as set by line 721, then $P_T$ should be $0$.
- Throughout Section 4.2: the first sentence of this section claims that it needs $\bar{\lambda} = \max_t \lambda_t$ to be known. However, I cannot find where $\bar{\lambda}$ is used in this section. The length of the blocks (Equations 6 and 7) does not use $\bar{\lambda}$. The value of $\rho$ in line 294 does not use it. Algorithm 1 also does not use it. In Theorem 2, the condition $\bar{\lambda} \leq \lambda_{\max}$ should be trivial from the problem setup, where $\lambda_t \leq \lambda_{\max}$ for all $t$. Can the authors clarify where $\bar{\lambda}$ is used?

*Theorem 4*:
- Again, in line 896, you are competing with a single best expert $j^*$. Each expert recommends a single threshold, therefore this implies that a single threshold, instead of a sequence of thresholds, is sufficient to obtain optimal guarantees. I find this to be contradictory to the claim in line 65 that a sequence of pricing thresholds is needed. Can the authors clarify on this? Intuitively, due to the adversarial nature of the sequence of jobs, it appears that a sequence of pricing thresholds must be needed and should do much better than a single threshold, but it seems that your results and their proofs show that a single threshold is sufficient. I find this very surprising.

*Sloppy writing:*
- Line 68 - 71 and everywhere else in the paper: the use of subscript "max'' is confusing, as it seems to denote both "the maximum values over $T$ rounds" like $x_{\max} = \max_{t = 1, 2, \dots, T} x_t$ and "the upper bound of the range of values" like $\lambda_t \in [\lambda_{\min}, \lambda_{\max}]$.
- Line 193: $\bar{\nu}_t$ is not defined until line 221.
- Line 221: should be $\mu_\tau$ instead of $\mu_t$
- Line 4 in Algorithm 1: in the argmax, it should specify that a \in {0, 1}.
- Theorem 1: $\nu$ is undefined. From the proof, it should be $\max_{\nu \in V}R(T, \nu)$, where $V$ is a bounded and convex set for the range of $\nu$. It is important that this set $V$ must be explicitly defined.
- Theorem 4: there should be a pseudo-code of the algorithm mentioned in this Theorem 4, at least in the appendix.
- Line 626: this line should go to the beginning of the appendix. As it is written, you are using $f_m$ everywhere without defining it. Also, I do not see the point of reducing $f_{\max}$  to $f_m$ "for simplicity". It makes the writing much more confusing without any benefits.

**Questions:**

Please address the concerns and questions raised in the Weaknesses above. Additional comments are below:

Because the techniques in the paper seem quite simple, and because I don't work on this problem setting directly, I couldn't tell whether the approaches are fundamentally new or not. I would be happy to raise the score if other reviewers and AC confirm that the results and proposed approaches are indeed novel, and that the authors' give clear answers to my concerns above.

---

> ### Author Response · Authors · 2025-11-21
>
> > Applying a regret bound to OLG-ALG without specifying the domain of $\nu$: assume $\nu \in \mathcal{V}$. Online learning algorithms generally require $\mathcal{V}$ to be bounded and convex; however, this set $\mathcal{V}$ is not defined anywhere.
>
> We thank the reviewer for pointing out this omission. In Appendix B (Line 616), we explicitly define the range of $\nu$ and show that the iterates $\nu_t$ generated by OLG-ALG remain in this set. In the revised version, we will move the definition and properties of $\mathcal{V}$ from the appendix to the main text where OLG-ALG is introduced.
>
>
> > Lacking a quantifier for $R(T,\nu)$: in many places the regret bound $R(T,\nu)$ is used without specifying $\nu$ or its domain. Can any $\nu$ be used here, or should it be $\max_{\nu \in \mathcal{V}} R(T,\nu)$? It likely should be $\max_{\nu \in \mathcal{V}} R(T,\nu)$ everywhere, since a specific $\nu$ is substituted in the bound (see line 685).
>
> We thank the reviewer for this helpful clarification request. In our analysis, the regret bound $R(T,\nu)$ holds for any fixed comparator $\nu$, and the online learning subroutine provides a uniform upper bound over all such $\nu$. As a result, the quantity can indeed be expressed as $\max_{\nu \in \mathcal{V}} R(T, \nu)$, as you suggested. In particular, at line 685 we choose a specific $\nu$ in order to control the term $\sum_{t \notin \mathcal{V}} \nu (\rho - a_t x_t) + R(T,\nu)$ and derive a clean upper bound. Because the regret inequality holds for every $\nu$, this substitution is valid even if we do not explicitly write $\text{max}_{\nu} R(T,\nu)$. We will clarify this point in our revision.
>
> > In line 65, the paper claims to adaptively learn and use a sequence $(\nu_t)_t$, which conflicts with the non-reusable setting that uses a single $\nu^\star$. However, if the appendix proofs are interpreted correctly, the regret bounds are actually applied with respect to a single $\nu$. For example, in Algorithm 1, the stated regret bound is a static bound for a single $\nu$.
>
> In the setting considered in Theorem 1, we set the average budget conservatively as $\rho = B/T$, and a single static comparator $\nu$ is sufficient to establish the performance guarantee. And this bound is optimal in the sense that it matches the lower bound in the worst-case regime $\lambda_t = T$. We will revise Line 65 to specific clarify Theorem 2 and Theorem 3's dynamic regrets are against dynamic comparators $\{\nu_t^\star\}_t$.
>
> > Line 269: writing $R(T,\nu) \le O(\sqrt{T})$ is incomplete; a factor capturing the loss scale is missing.
>
> We thank the reviewer for pointing this out. Since we treat the online learning subroutine OL-ALG in a black-box manner, we omitted constant factors in the main text for simplicity because it is not the main focus of our paper. In Appendix B (Line 611), we explicitly bound both the iterates $\nu_t$ and the gradients $g_t$ (often required in regret bounds), which in turn determine the loss scale in the regret bound, i.e., $R(T,P_T) \leq O\left(GD\sqrt{T(1+P_T)}\right)$, where $G=\max\{\rho,\, x_{\text{max}} - \rho\}, D= \frac{f_{\text{max}}}{x_{\min}}$. We will explicitly clarify this in our revision.
>
> > Continuing the point above, in the proof of Theorem 2, despite the theorem using a path-length bound, all comparators in active blocks $\nu_t^\star$ are set to be equal to $f_m/x_m$ (line 721). Does this not defeat the purpose of using a dynamic regret bound?
>
> $\nu_t^\star$ is set to $0$ when the constraint-violation set $\mathcal{V}_i$ is empty (Line 709), and it is set to $f_m/x_m$ when $\mathcal{V}_i$ is non-empty. Our goal is to choose $\nu_t^\star$ so as to control and upper bound the term
>
> $$
> \sum_{t \notin \mathcal V_i} \nu_t^* (\rho - a_t x_t)
> +
> \sum_{t \in \mathcal V_i} \left( w_t(\nu_t) - w_t(\nu_t^*) \right).
> $$
>
> Although within each active block $\nu_t^\star$ is constant, it changes across blocks depending on whether the constraint is violated, and the path-length term quantifies the cost of these changes. Therefore, the use of a dynamic regret bound is not vacuous here.
>
> > In Theorem 2, the condition $\bar{\lambda} \le \lambda_{\max}$ should already follow from the setup, where $\lambda_t \le \lambda_{\max}$ for all $t$. Can the authors clarify where $\bar{\lambda}$ is used and why it needs to be stated separately?
>
> In our notation, $\bar{\lambda} = \max_{t \in [T]} \lambda_t$ is the (unknown) true maximum duration of jobs in the instance, whereas $\lambda_{\max}$ is a platform-chosen parameter of the algorithm. In practice, the platform may not know $\bar{\lambda}$ exactly, but it can typically specify a conservative upper bound $\lambda_{\max}$ (see Lines 71 and 230), for example via a system-imposed cap on the declared job duration. Moreover, since $\lambda_{\max}$ is manually set, it can be tuned to trade off tightness of the bound, as discussed around Line 324.

---

> ### Author Response · Authors · 2025-11-21
>
> > Again, in line 896, you are competing with a single best expert $j^\star$. Each expert recommends a single threshold, so this implies that a single threshold, rather than a sequence of thresholds, suffices to obtain optimal guarantees. This seems to contradict the claim in line 65 that a sequence of pricing thresholds is needed. Can the authors clarify this?
>
> We thank the reviewer for raising this point. Our claim in line 65 refers specifically to the primal–dual based dynamic pricing scheme. By contrast, the expert-based construction analyzed around line 896 has a different structure: each expert corresponds to a *fixed two-sided* threshold (i.e., an interval) rather than a single one-sided threshold as in the primal–dual method. We prove there exists a fixed interval of thresholds that achieve $\alpha$ competitive with the specific $\alpha$.
>
> > Because the techniques in the paper seem quite simple, and because I don't work on this problem setting directly, I couldn't tell whether the approaches are fundamentally new or not.
>
>  To the best of our knowledge, our work is the first to study online reusable resource allocation with adversarial arrivals and budget (capacity) constraints, which is a fundamentally challenging setting. In particular, this setting is more challenging than the relatively well studied special cases with non-reusable resources or in stochastic environments.
>
> Our contributions: First, we develop a dynamic online-learning-based framework specifically tailored to reusable resources, which differs from the classical primal–dual and static-threshold approaches studied in non-reusable allocation problems. Second, we introduce and analyze *two-sided* threshold structures that arise naturally in the reusable-resource model. We also consider a flexible allocation setting, and give a near optimal reduction matches the lower bound. These elements, together with the accompanying competitive and regret analyses, require new structural arguments specific to the reusable, adversarial requests and are, in our view, non-trivial and novel extensions of prior primal–dual and online learning approaches. We will revise the technical overview to incorporate this discussion.

---

> > ### Comment · Reviewer_tJEB · 2025-11-27
> >
> > Thanks for the clarification. Could you upload a revision with the aforementioned corrections? It is not easy for me to evaluate whether your answers look good or not without examining the new bounds explicitly written out.

---

> ### Author Response · Authors · 2025-11-28
>
> We appreciate the reviewer’s suggestion. We have uploaded an updated version of the paper. The changes regarding the regret bounds only introduce constant factors into the regret term of the online learning subroutine OL-ALG and do not change the order of the regret or any of the main competitive-ratio guarantees.

---

### Official Review · Reviewer_2mRk · 2025-11-02

**Soundness:** 3
**Presentation:** 3
**Contribution:** 3
**Rating:** 6
**Confidence:** 3

**Summary:**

The paper tackles online allocation with reusable capacity under adversarial arrivals, where each job ties up part of a budget for a duration and yields a revenue. It proposes two families of algorithms: (i) primal–dual dynamic pricing (variants alternating-block and duration-aware), and (ii) a pricing-experts approach that learns a threshold over time. Theoretical results give asymptotic competitive-ratio and dynamic-regret guarantees; for a flexible-allocation extension, a powers-of-two reduction offers a ~1/log-B style approximation with a near-matching lower bound. The work positions itself as a first systematic adversarial treatment of reusable resources with ties to GPU/cloud scheduling.

**Strengths:**

- Clear problem formulation that unifies multiple adversarial reusable-resource scenarios with a dynamic-price view
- Two complementary solution routes (primal–dual and experts) with nontrivial analyses, including a duration-aware variant that removes alternating blocks in the constant-duration regime.
- Flexible-allocation extension with a simple discretization and a near-matching lower bound that clarifies the cost of reduction

**Weaknesses:**

- I find the strongest guarantees hard to map to practice. Several results still scale with $\lambda_{\max} x_{\max} / B$ (and the alternating-block scheme effectively halves active time), while additive terms and dynamic-regret/path-length dependencies can dominate unless horizons and capacities sit in a favorable regime. I would recommend the authors to pin down concrete parameter ranges (e.g., GPU-cluster scales) where the bounds are $<$1 and additive terms are provably negligible relative to OPT
- I’m concerned about reliance on parameters and structural assumptions: bounds often need a known $\lambda_{\max}$ (sometimes constant), constraints such as $x_{\max}\le B/2$ or $B/3$, and even revelation of $\lambda_t$ at arrival; the flexible case assumes monotone $\lambda_t(x)$. My confidence would improve with robustness analyses (e.g., hedging or doubling over misspecified $\lambda_{\max}$, handling noisy/unknown durations) and a brief comparison to stochastic/replenishable models to clarify when these assumptions are realistic.

**Questions:**

Please address the concerns raised above.

---

> ### Author Response · Authors · 2025-11-21
>
> > I find the strongest guarantees hard to map to practice. Several results still scale with $\lambda_{\text{max}} x_{\text{max}} / B$, while additive terms and dynamic-regret/path-length dependencies can dominate unless horizons and capacities sit in a favorable regime. I would recommend the authors to pin down concrete parameter ranges (e.g., GPU-cluster scales) where the bounds are < 1 and additive terms are provably negligible relative to OPT.
>
> We thank the reviewer for this insightful comment and for encouraging us to clarify the practically relevant parameter regimes for our guarantees. Our main competitive-ratio bound, $x_{\text{max}} / (B / \lambda_{\text{max}})$, strictly improves upon the previous guarantee $x_{\text{max}} / (B / T)$ and is designed to capture realistic cluster scales.
>
> Consider the following real-world scenario, in a typical multi-tenant hyperscale cluster with $B \approx 60{,}000$–$100{,}000$ GPUs, most teams run jobs with at most $x_t \le 64$–$512$ GPUs for durations $\lambda_t \le 2$ days. If we take one-hour slots (meaning that the minimum duration to rent a GPU is 1 hour), this corresponds to $\bar{\lambda} = 48$ hours and we may set $\lambda_{\text{max}} = 100$ one-hour slots. Under these parameters, $\lambda_{\text{max}} x_{\text{max}} / B$ is well below $1$ (e.g., on the order of $0.06$–$0.85$), so our competitive-ratio factor lies in the desired regime. If one instead uses one-day slots, the factor becomes even smaller. We will add these concrete ranges and a short discussion of representative GPU-cluster scales in the revision.
>
> Regarding the additive terms: the regret and path-length terms in our bounds are sublinear in the horizon in all cases, so they are never dominate asymptotically. For example, for a horizon of $T = 10{,}000$ one-hour slots, the regret term is on the order of $10^2$, which is about $1\%$ of the cumulative revenue from $10{,}000$ jobs, and hence negligible relative to $\mathrm{OPT}$. Consequently, the algorithm achieves no-regret revenue with respect to $\frac{1}{2\alpha}\mathrm{OPT}_b$ and is essentially $\frac{1}{2\alpha}$-competitive.
>
> Finally, our framework does not hinge on a single “strongest” guarantee: practitioners can select, among our algorithms, the one whose assumptions on $(\bar{\lambda}, x_{\text{max}}, B)$ and arrival structure best match their deployment setting. In fact, they can even experiment with our algorithms and choose the one with the best empirical performance. We will make this discussion clearer in our revision.
>
> > I’m concerned about reliance on parameters and structural assumptions: bounds often need a known $\lambda_{\text{max}}$, constraints such as $x_{\text{max}}\leq B/2$, and even revelation of $\lambda_t$ at arrival. The flexible case assumes monotone $\lambda_t(x)$.
>
> We thank the reviewer for raising these concerns. Importantly, all of the parameters and structural assumptions we use correspond to standard mechanisms that real-world platforms already expose or can enforce through simple configuration.
>
> - $\lambda_{\text{max}}$ only needs to be a rough upper bound on the true $\bar{\lambda} = \text{max}_{t\in[T]} \lambda_t$, and our framework allows the platform to select different algorithms based on the available knowledge of $\bar{\lambda}$ (see Line 074 and Line 231). And in practice, batch and cloud schedulers routinely require each job to declare a maximum runtime at submission, and jobs that exceed this limit are rejected or terminated.
>
> - Constraints such as $x_{\text{max}}\leq B/2$ or $x_{\text{max}}\leq B/3$ are also realistic, because individual jobs typically occupy only a small fraction of the cluster. For instance, in the GPU-cluster example discussed in our previous response, we have $B \approx 60{,}000$–$100{,}000$ and $x_t \le 64$–$512$, and it's hard to imagine a single job demanding 20000 GPUs in any case, so these conditions are comfortably satisfied.
>
> - The monotonicity assumption on $\lambda_t(x)$ is natural in this setting: if a job is allocated more GPUs, it should complete faster, or at least no slower, than with fewer GPUs. This assumption is standard in performance modeling and aligns with how practitioners reason about scaling compute resources.
>
> We will clarify these points and better emphasize the practical implementability of these assumptions in the revised version.

---

> ### Author Response · Authors · 2025-11-21
>
> > My confidence would improve with robustness analyses (e.g., hedging or doubling over misspecified $\lambda_{\text{max}}$, handling noisy/unknown durations).
>
> We thank the reviewer for this insightful suggestion. Regarding the duration assumptions, in many practical platforms the operator can impose an upper bound or cap on the service duration (as in the examples discussed above), so our assumption of a capped maximum duration is realistic and implementable in real-world systems.
>
> We agree that incorporating robustness analyses—such as hedging against misspecified $\lambda_{\text{max}}$ and explicitly handling noisy or unknown durations—is an important and promising research direction. However, even in the current setting the problem is already technically challenging and non-trivial, so in this work we deliberately focus on the fundamental version of the model in order to isolate and resolve the core algorithmic and structural difficulties. Extending our framework to explicitly model noisy or misspecified durations is beyond the scope of the present paper but represents a natural direction for future work.

---

### Author Response · Authors · 2025-12-04

Dear AC,

Thanks for taking the time to review our work. We believe we have addressed the main concerns raised by reviewers and and have uploaded an updated PDF incorporating their suggestions. Below we briefly summarize the key points of our responses to each reviewer:


---

### Reviewer 2mRk

- **Practical relevance of bounds.** We now provide concrete GPU-cluster scales under which $\lambda_{\max} x_{\max} / B < 1$, and we show that the regret terms are at most about $1\%$ of the revenue for realistic horizons, so the guarantees are meaningful in practice.
- **Assumptions on $\lambda_{\max}, x_{\max}, \lambda_t(x)$.** We clarify that these are standard and implementable in practice (e.g., duration caps, typical job sizes, and the natural monotonicity of $\lambda_t(x)$).

---

### Reviewer tJEB

Many of reviewer tJEB’s concerns are about notations and definitions; we have made these explicit in the revised version.

- **Range of $\nu$ and quantification of $R(T,\nu)$.**
   We explicitly define the range of the dual variable in the main text (previously only in the appendix) and clarify that the regret bound holds uniformly for any fixed comparator $\nu$. We now write this as $ \max_{\nu} R(T,\nu)$ in our revision.

- **Use of dynamic regret in Theorem 2.**
   We clarify the role of dynamic regret in Theorem 2: the comparator sequence $\{\nu_t^\star\}$ we construct is block-wise and changes across blocks, and the path-length term captures the cost of these changes. So the dynamic regret bound is genuinely used in the analysis.

- **Loss-scale constant in $O(\sqrt{T})$.**
   In the original submission we omitted the constant factor related to the loss scale in the main text for simplicity. In the revision we make this explicit, writing the regret as $O(GD\sqrt{T})$. These changes introduce only constant factors into the regret term of OL-ALG and do not affect the order of the regret or any competitive ratio guarantees.

---

### Reviewer 1QE6

- We expanded the related work section with a more detailed discussion of online packing / online LP.

---

### Meta-Review · Area_Chair_ZrjL · 2026-01-02

**Summary:**

The Reviewers are mixed about this paper, with two of them mildly supporting acceptance (score 6) and one of them supporting rejection (score 4). The main concerns raised by the Reviewers are: (i) the guarantees obtained by the algorithms proposed in the paper are hard to map in practical settings, (ii) the results depend on some parameters that induce trade-offs that are not fully understood, and, thus, it is not clear how one should manage such parameters in practice, and (iii) the writing of the paper is generally not adequate and there are multiple small errors in the proofs as well. Despite all these concerns, the Reviewers still acknowledge that the problem studied in the paper is interesting and that there is some degree of novelty (though maybe limited) in the results presented in the paper.

Given the considerations above, I think that the paper is borderline, as I do **not** see neither any strong reason to accept it nor a clear motivation to reject it. Nevertheless, I think that, given the initial scores received by the paper, acceptance would have required some of the  Reviewers to raise their score in the discussion period. By reading the rebuttals posted by the Authors, I do **not** think that some of the Reviewers would have changed their score. Indeed, the responses to the weaknesses (i) and (ii) above are not fully convincing, and I think that the Reviewers would have not changed their scores based on those. Thus, my conclusion is that the paper does **not** pass the bar for acceptance at ICLR at this stage. I strongly encourage the Authors to work more on the issues above in the future versions of the paper, as it has potential for being a solid paper.

**Reviewer Concerns:**

The main concerns raised by the Reviewers are: (i) the guarantees obtained by the algorithms proposed in the paper are hard to map in practical settings, (ii) the results depend on some parameters that induce trade-offs that are not fully understood, and, thus, it is not clear how one should manage such parameters in practice, and (iii) the writing of the paper is generally not adequate and there are multiple small errors in the proofs as well. Despite all these concerns, the Reviewers still acknowledge that the problem studied in the paper is interesting and that there is some degree of novelty (though maybe limited) in the results presented in the paper.

The responses provided in the rebuttals to the weaknesses (i) and (ii) above are not fully convincing, and I think that the Reviewers would have not changed their scores based on those. Specifically, the motivating example provided by the Authors is **not** sufficiently convincing.

**Reviewer Scores:**

Reviewer 2mRk, Score: 6 - I believe that the rebuttal would not have changed the reviewer’s opinion, especially given the other reviews.

Reviewer tJEB, Score: 4 - I believe that the rebuttal would not have changed the reviewer’s opinion, especially given the other reviews.

Reviewer 1QE6, Score: 6 - I believe that the rebuttal would not have changed the reviewer’s opinion, especially given the other reviews.

---

### Decision · Program_Chairs · 2026-01-26

Reject